# Anomalies of Coset Non-Invertible Symmetries

Po-Shen Hsin[1], Ryohei Kobayashi[2], Carolyn Zhang[3]

[1] *Department of Mathematics, King's College London, Strand, London WC2R 2LS, UK*

[2] *School of Natural Sciences, Institute for Advanced Study, Princeton, NJ 08540, USA*

[3] *Department of Physics, Harvard University, Cambridge, MA02138, USA*

## Abstract

Anomalies of global symmetries provide important information on the quantum dynamics. We show the dynamical constraints can be organized into three classes: genuine anomalies, fractional topological responses, and integer responses that can be realized in symmetry-protected topological (SPT) phases. Coset symmetry can be present in many physical systems including quantum spin liquids, and the coset symmetry can be a non-invertible symmetry. We introduce twists in coset symmetries, which modify the fusion rules and the generalized Frobenius-Schur indicators. We call such coset symmetries twisted coset symmetries, and they are labeled by the quadruple $(G, K, \omega_{D+1}, \alpha_D)$ in $D$ spacetime dimensions where $G$ is a group and $K \subset G$ is a discrete subgroup, $\omega_{D+1}$ is a $(D+1)$-cocycle for group $G$, and $\alpha_D$ is a $D$-cochain for group $K$. We present several examples with twisted coset symmetries using lattice models and field theory, including both gapped and gapless systems (such as gapless symmetry-protected topological phases). We investigate the anomalies of general twisted coset symmetry, which presents obstructions to realizing the coset symmetry in (gapped) symmetry-protected topological phases. We show that finite coset symmetry $G/K$ becomes anomalous when $G$ cannot be expressed as the bicrossed product $G = H \bowtie K$, and such anomalous coset symmetry leads to symmetry-enforced gaplessness in generic spacetime dimensions. We illustrate examples of anomalous coset symmetries with $A_5/\mathbb{Z}_2$ symmetry, with realizations in lattice models.

November 5, 2025

# 1 Introduction

In recent years, generalization of group-like symmetry, called non-invertible symmetry, has developed rapidly with many examples in field theories and lattice models (see e.g. [1, 2, 3, 4, 5, 6, 7, 8, 9]). Non-invertible symmetry differs from group-like symmetry in that their fusion rules are not invertible. The symmetry operators do not obey a group multiplication law, and instead form a (higher) fusion category. While there is much progress in understanding mathematical properties of non-invertible symmetries, they often require sophisticated techniques in fusion categories which could be challenging to compute explicitly.

In this work, we will study a class of non-invertible symmetries that come from cosets $G/K$ where $K \subset G$ is a not necessarily a normal subgroup of $G$, continuing our previous work [10] (see also [11, 12, 13, 14, 15, 16, 17, 18, 19] for other examples). The coset symmetry is obtained by starting with a theory with $G$ symmetry and gauging the possibly non-normal subgroup $K$. The coset symmetry defects can be constructed using a sandwich construction. Inside the sandwich we have a $G$ symmetry defect (possibly with 't Hooft anomalies), and outside the sandwich the non-anomalous $K$ subgroup is gauged, possibly with a topological action, and the interfaces on the two sides correspond to Dirichlet boundary condition of $K$ gauge field (see Fig. 1).

In this definition, the coset symmetry comes with $\mathrm{Rep}(K)$ symmetry– they are generated by the Wilson lines of $K$ gauge field. In particular, this means that although as cosets $G'/K' = G/K$ for $K' = K/N, G' = G/N$ for common normal subgroup $N$ of $G, K$, the symmetries $G/K$ and $G'/K'$ differ by $\mathrm{Rep}(K), \mathrm{Rep}(K')$. The groups $G, K$ depend on the spectrum of topological lines – if they form $\mathrm{Rep}(K)$, then the pair of groups are $(G, K \subset G)$ such that $G/K$ equals the coset. In particular, if the coset is a group and there are no topological line operators, then $K = 1$. If the coset is not a group, then there must be topological line operators. We will discuss more in the detail the definition of coset symmetry in Sec. 2. In particular, we will introduce suitable twists for the symmetry that can change the fusion rules and higher group structures of the coset symmetry.

Coset symmetry has been found in many physical contexts. For example, Alice electrodynamics at low energy has $O(2)/\mathbb{Z}_2^{\mathcal{C}}$ symmetry, where the charge conjugation symmetry is gauged. In such systems, there are Alice rings– the twist defects for charge conjugation

symmetry, which have been observed in experiments such as the recent spin-1 Bose Einstein condensation for 23Na or radioactive 87Rb atoms [20, 21]. Thus the presence of dynamical Alice rings (i.e. twisted defects for charge conjugation), such as those formed from dynamically unstable monopoles, is an experiment signature for the coset symmetry.

We will investigate the anomalies of coset symmetries. Anomalies are important properties of global symmetries, and we will focus on the obstruction to symmetry-protected topological phases, i.e. a gapped systems with a unique, symmetric ground state on any manifold. This is important in constraining the dynamics: this type of anomaly forces the low energy physics to be nontrivial. There is another definition of anomaly in the literature as an obstruction to gauging the symmetry, but we will not discuss it in detail here (see Sec. 5 for more comments).

An important tool for understanding these anomalies is the bulk topological quantum field theory (TQFT) in one higher dimension that describes the symmetry defects [22, 23, 24, 25, 26, 27]. That is, the theory in $D$ dimension is considered as a bulk TQFT in $(D + 1)$ dimension on a thin interval, sandwiched by the boundary conditions. The global symmetry of the theory is then realized by topological operators of the gapped boundary at one end of the interval. If the bulk TQFT admits another gapped boundary whose nontrivial boundary topological excitations do not overlap with those of the symmetric gapped boundary, then the bulk TQFT reduces on an interval with the boundary conditions provides a trivially gapped phase with the symmetry – this means that there is no first type of anomaly for the symmetry. Such methods have been used to rule out trivially gapped phase with non-invertible symmetries in [6, 7].

In this work, we will explore the constraints on the low energy spectrum of the system enforced by anomalies of coset symmetries.

## 1.1 Obstruction to SPT phases: three situations

Let us elaborate more on the obstruction to SPT phases. For (internal) invertible symmetries, if we only specify the fusion rules without any further information, then for a given set of fusion rules there are non-anomalous invertible symmetries that can be realized in SPT phases and such symmetries also can be gauged. If the non-anomalous symmetries carry nontrivial topological response, then the invertible symmetries with non-quantized responses cannot be realized in SPT phases. For example, if we specify the $U(1)$ fusion rule for 0-form symmetry and also fractional Hall response in 2+1D, such symmetry cannot be realized in SPT phases since the response is not integer, though the $U(1)$ symmetry can still be gauged (see e.g. [28, 29]).

When the symmetry is invertible, the obstruction to SPT phases fall into the following categories:

(1) The symmetry has an 't Hooft anomaly, described by nontrivial bulk invertible phase with topological action $\omega_{D+1} \neq d\beta_D$ (here $D$ is the spacetime dimension). In particular,

the bulk topological term cannot be continuously tuned to zero without breaking the relevant symmetries. Such symmetry cannot be realized in SPT phases.

(2) The symmetry does not have an 't Hooft anomaly, but it has a fractional topological response. This is described by $\omega_{D+1} = d\beta_D \neq 0$. $\beta_D$ is the fractional response. The bulk term is a boundary term that is not well-defined by itself without a bulk. Such symmetry cannot be realized in SPT phases.

(3) The symmetry does not have 't Hooft anomalies, and also does not have fractional topological response $\omega_{D+1} = 0$. Such symmetry can be realized in SPT phases.

We remark that the fractional/integer response can be captured by the symmetry defect configurations where the defects intersect. For example, if the defect locally can be described by a current, the response is given by the parity-odd contact terms in the current operator product expansion: in 2+1D, it is given by [10, 30]

$$J_\mu(x)J_\nu(0) \supset \frac{i\sigma_H}{2\pi}\epsilon_{\mu\nu\lambda}\partial^\lambda\delta^3(x) \ , \tag{1.1}$$

where $\sigma_H$ is the response coefficient and this is in Euclidean spacetime.

While the bulk TQFT understanding of the situations (1),(3) is reasonably well, the bulk TQFT understanding of situation (2) is less useful. In particular, a fractional response can be any exact bulk cocycle, i.e., group cochain in $D$ dimension.

When the symmetry is non-invertible, in general it is an open problem how to distinguish the categories, in particular separate the categories (1),(3) from (2). Here, we will focus on the simpler situation of coset symmetry, where the bulk TQFT is still a gauge theory for ordinary groups, and we can still use the above distinctions for the three categories.

## 1.2 Summary of results

We show a general coset symmetry in $D$ spacetime dimensions can be described by the quadruple $(G, K, \omega_{D+1}, \alpha_D)$, where $K$ is a subgroup of $G$, $\omega_{D+1}$ is a $(D+1)$-cocycle for group $G$ with $U(1)$ coefficient, such that $\omega_{D+1}|_K = d\alpha_D$. Such symmetry is obtained by starting with invertible $G$ symmetry with an 't Hooft anomaly $\omega_{D+1}$, and then gauging a $K$ subgroup symmetry with the twist $\omega_D$. This is a generalization of the group theoretical fusion category for $D = 2$ [31] and the group theoretical fusion 2-category for $D = 3$ [32] to generic spacetime dimensions. The twists have the following consequences on the algebraic structure of the coset symmetry:

- The twist $\omega_{D+1}$ can modify the fusion rules and associator of the symmetry.
- The twist $\alpha_D$ can modify the fusion rules of the symmetry.

When both $\omega_{D+1} = 0, \alpha_D = 0$, we call the coset symmetry an untwisted coset symmetry.

In our discussion, we include the fractional topological response as part of the symmetry data similar to how systems with Lieb-Schultz-Mattis (LSM) anomalies from translation and filling constraints, one also has to specify the filling [33, 34]. The filling is not a piece of data from the symmetry algebra and indeed does not appear in the symTFT construction, but can still impose important constraints on dynamics. It would be interesting to understand how to incorporate this data into the symTFT description.

We note that the twist $\omega_{D+1}$ constrains the possible coset symmetries: it rules out the subgroup $K$, along with the symmetry lines $\text{Rep}(K)$, if $\omega_{D+1}|_K$ is not exact.

In addition, the presentation of the coset symmetry $(G, K, \omega_{D+1}, \alpha_D)$ is in general redundant: there can be two presentations describe the same symmetry. For instance, this can arise when the bulk TQFT admits two (or more) descriptions: one as $G$ gauge theory with topological action $\omega_{D+1}$, and another as $G' \neq G$ gauge theory with topological action $\omega'_{D+1}$.

In the following, let us summarize the physical consequences of the coset symmetry in general quantum systems.

**Dynamical consequence of anomalies**   We show that the finite coset symmetry leads to the following constraint on the dynamics:

- If a finite group $G = H \bowtie K$ is a bicrossed product of $K$ with another group $H$, then the coset symmetry in $D \geq 3$ can be realized in symmetric gapped phases. [1] Furthermore, if one can find a choice of the subgroup $H \subset G$ in the expression $G = H \bowtie K$ such that the twist $\omega_{D+1}$ restricted to $H$ becomes trivial, i.e., $[\omega_{D+1}|_H] = 0$ in $H^{D+1}(BH, U(1))$, then the coset symmetry is anomaly free and can be realized in SPT phases.

- If a finite group $G$ cannot be expressed as a bicrossed product $G = H \bowtie K$ with any subgroup $H \subset G$, then the coset symmetry is anomalous and exhibits symmetry-enforced gaplessness. That is, if the system preserves both the $\text{Rep}(K)$ symmetry of Wilson lines and the 0-form coset non-invertible symmetry, the system must be gapless.

For instance, the coset non-invertible symmetries $A_5/\mathbb{Z}_2$, $A_6/A_5$ do not admit the expression in terms of bicrossed product, and these coset symmetry with or without twist give examples of anomalous coset symmetries that lead to symmetry-enforced gaplessness. Meanwhile, generic finite coset symmetry can be realized in $K$ gauge theory with spontaneously broken symmetries, which is a gapped phase that spontaneously breaks the $\text{Rep}(K)$ symmetry of Wilson lines.

We also discuss dynamical scenarios for continuous coset symmetries that cannot be realized by SPT phases. We show that continuous coset symmetry can enforce the dynamics

---

[1]For two subgroups $H, K$ of $G$, $G$ is the bicrossed product $G = H \bowtie K$ if and only if $G$ can be expressed as $G = HK$ and $H \cap K = \{\text{id}\}$. Equivalently, each group element $g \in G$ has a unique expression as $g = hk$ with $h \in H, k \in K$.

to be gapless, such as the coset symmetry $G = SU(2)$, $K =$ finite subgroup of $SU(2)$, and $\omega_{D+1}$ given by Witten anomaly in $D = 4$ spacetime dimensions.

Any system with coset symmetry with anomalies or fractional responses must flow to nontrivial phases. We present examples of systems with fractional responses of coset symmetry that originate from nontrivial twists $\omega_{D+1}, \alpha_D$. For example, the twisted coset symmetry in two 2+1D massless Dirac fermions coupled to $\mathbb{Z}_2$ gauge field can be enriched with twisted coset symmetry $O(2)/\mathbb{Z}_2$, with $\omega$ being the theta term corresponds to the $O(2)_{1/2,1/2}$ fractional Chern-Simons term and $\alpha = \frac{1}{2}\eta$ being half of the minimal $\mathbb{Z}_2$ Chern-Simons term, where the notation is the same as [35]. Such twisted coset symmetry exhibits fractional response and cannot be realized in SPT phases.

Systems with finite $G/K$ coset symmetries can be realized in lattice models of $K$ gauge theory. We discuss two examples of lattice gauge theories: $K$ gauge theory with untwisted coset symmetry $(H \bowtie K)/K$ (without anomaly), and $\mathbb{Z}_2$ gauge theory with the untwisted coset symmetry $A_5/\mathbb{Z}_2$ (with anomaly). We demonstrate that one can condense electric charges of $K$ gauge theory while preserving $(H \bowtie K)/K$ symmetry, which leads to the Higgs phase of $K$ gauge theory. The Higgs phase is regarded as an SPT phase with $(H \bowtie K)/K$ symmetry, being consistent with the fact that untwisted $(H \bowtie K)/K$ symmetry is anomaly free. In contrast, we demonstrate that the coset $A_5/\mathbb{Z}_2$ symmetry forbids the condensation of electric charges in $\mathbb{Z}_2$ gauge theory; the anomaly of $A_5/\mathbb{Z}_2$ symmetry gives the obstruction to the Higgs phase preserving the coset symmetry. In a companion paper [36], we will discuss a larger class of lattice gauge theory models with coset symmetry realized in Higgs phases, and explore their responses and anomalies.

The rest of the paper is organized as follows. In section 2, we introduce twisted coset symmetry that generalizes the coset symmetry labeled by $G$ and subgroup $K$. In section 3 we discuss the anomalies of twisted coset symmetries as obstructions to their realization in SPT phases. In section 4 we construct explicit lattice models with anomalous coset symmetries. In section 5 we comment on the obstruction to gauging the twisted coset symmetry, as well as discuss several future directions. There are two appendices: in appendix A we discuss the anomaly-free condition for coset symmetries with $G$ given by bicrossed product, and in appendix B we discuss an example of anomalous coset symmetry $A_6/A_5$.

# 2 Twisted Coset Non-Invertible Symmetries

## 2.1 Definition of twisted coset symmetry: $(G, K, \omega_{D+1}, \alpha_D)$

In this section, we will generalize the definition of coset symmetry in [10] for spacetime dimension $D$, into the following data $(G, K, \omega_{D+1}, \alpha_D)$ that describes gauging a subgroup $K \subset G$ in invertible $G$ symmetry:

- A group $G$ and a finite subgroup $K \subset G$. They give the coset $G/K$. For a given coset,

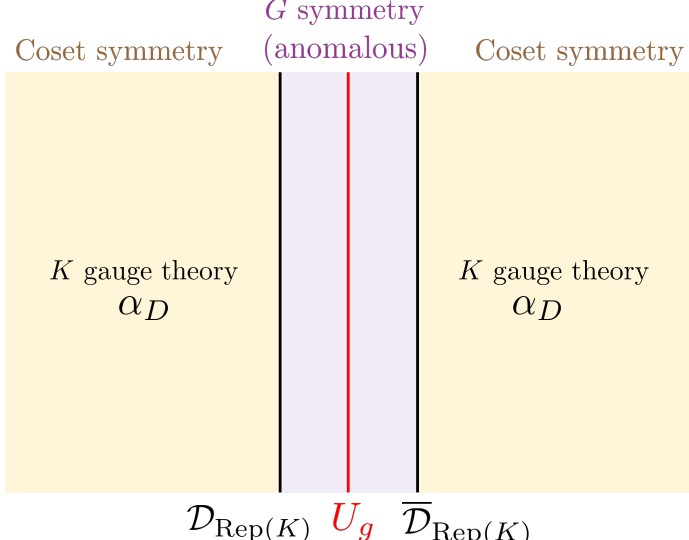

Figure 1: Sandwich construction for twisted coset symmetry defects in $D$ spacetime dimensions. The middle region has anomalous $G$ symmetry defect, with anomaly given by cocycle $\omega_{D+1}$. In the outer region on the two sides, the $K \subset G$ symmetry is gauged with topological action $\alpha_D$. The two interfaces separating the region is given by Dirichlet boundary condition of the $K$ gauge field. The subgroup $K$ is finite to ensure the defect is topological. The symmetry defect of the $K$ gauge theory has the form of the sandwich $\tilde{U}_g = \overline{\mathcal{D}}_{\text{Rep}(K)} \times U_g \times \mathcal{D}_{\text{Rep}(K)}$.

the groups $G, K$ are such that the topological line operators are $\text{Rep}(K)$. In particular, the coset symmetry has a subcategory given by $(D-1)\text{Rep}(K)$ that consists of the Wilson lines of $K$ and their condensation descendants. When $G$ and $K$ do not have a common normal subgroup $N$, the coset symmetry is said to be minimal.

- Topological action for $G$ in $D+1$ spacetime dimension, $\omega_{D+1} \in H^{D+1}(BG, U(1))$. This represents that $G$ symmetry has an 't Hooft anomaly characterized by bulk SPT response $\omega_{D+1}$.

- Topological action for $K$ in $D$ spacetime dimension, $\alpha_D \in C^D(BK, U(1))$. The subgroup $K$ needs to satisfy

$$\omega_{D+1}|_K = d\alpha_D , \tag{2.1}$$

where $\omega_{D+1}|_K$ is the pullback under the inclusion $K \to G$.[2] Different $\alpha_D$ are related by $\alpha_D \to \alpha_D + \eta_D$ with $\eta_D \in Z^D(BK, U(1))$. This represents that an anomaly free subgroup $K \subset G$ is gauged with the twist valued in $\alpha_D$, and the twist can be shifted by $D$-dimensional SPT response $\eta_D$. Such a shift in $\alpha_D$ may or may not change the symmetry category depending on the cohomology class $[\eta_D]$, as we will see in Sec. 2.1.1.

---

[2]A similar constraint is also imposed in the study of Cheshire strings in twisted gauge theories [37].

**"Sandwich construction" of twisted coset symmetry**   A large class of symmetry defects for the twisted or untwisted coset symmetries can be expressed in terms of a "sandwich construction" following the description in [10] (see Figure 1).

In the middle there is symmetry defect of the symmetry $G$, with 't Hooft anomaly described by cocycle $\omega_{D+1}$ for spacetime dimension $D$. On the two sides of the $G$ defect we place condensation defects of $K$ gauge theory, given by Dirichlet boundary conditions of the $K$ gauge fields. Outside the sandwich the $K$ symmetry is gauged with topological action $\alpha_D$. That is, the symmetry defect is expressed by the combination

$$\tilde{U}_g = \mathcal{D}_{\text{Rep}(K)} \times U_g \times \overline{\mathcal{D}}_{\text{Rep}(K)} \tag{2.2}$$

where $U_g$ is the $g \in G$ symmetry defect of the anomalous $G$ symmetry, and $\mathcal{D}_{\text{Rep}(K)}$ is a half-gauging defect of $K$ with the topological action $\alpha_D$. The half-gauging defects have the fusion rule

$$\overline{\mathcal{D}}_{\text{Rep}(K)} \times \mathcal{D}_{\text{Rep}(K)} = \sum_{k \in K} U_k, \quad \mathcal{D}_{\text{Rep}(K)} \times U_k = \mathcal{D}_{\text{Rep}(K)}, \quad U_k \times \overline{\mathcal{D}}_{\text{Rep}(K)} = \overline{\mathcal{D}}_{\text{Rep}(K)}. \tag{2.3}$$

It follows that the sandwich defects obey the fusion rule

$$\tilde{U}_g \times \tilde{U}_h = \sum_{k \in K} \tilde{U}_{gkhk^{-1}}. \tag{2.4}$$

One can see that the twist $\omega_{D+1}$ does not modify the fusion rules of the above symmetry defects $\{\tilde{U}_g\}$, since the twists do not affect the fusion rules of $U_g$ operators (only the associators of these operators are modified by the phase factors).

This topological defect $\tilde{U}_g$ becomes simple if and only if $K \cap gKg^{-1} = \{\text{id}\}$. We emphasize that the sandwich defects generally do not give simple topological operators. In particular, the operator $\tilde{U}_g$ in (2.2) is not simple for any $g \in G$ when the coset symmetry is not minimal, i.e., $G$ and $K$ shares a nontrivial common normal subgroup $N$. In that case, one can find a sandwiched topological defect $\tilde{U}'_g$ with smaller degeneracy as $\tilde{U}'_g = \mathcal{D}_{\text{Rep}(K/N)} U_{gN} \overline{\mathcal{D}}_{\text{Rep}(K/N)}$; this is obtained by first gauging $N$ symmetry of the theory and then considering the sandwich of $G/N$ symmetry defect by the half gauging of $K/N$. This implies that the above $\tilde{U}_g$ in (2.2) is not simple in such coset symmetry, but rather obtained after combining the smaller coset symmetry defect $\tilde{U}'_g$ with the condensation defect for $\text{Rep}(K)$ Wilson lines.

Although the fusion rule of the sandwiched defects is independent of the twist $\omega_{D+1}$, the fusion rules of the simple topological operators will be modified by the twist $\omega_{D+1}$ in general. In Sec. 2.2, we discuss the modified fusion rule of the twisted coset symmetry in detail.

### 2.1.1   Description using bulk TQFT

We can describe the twisted coset symmetry using a bulk symmetry TQFT with a gapped boundary, given by $G$ gauge theory in $(D+1)$ spacetime dimensions with topological action

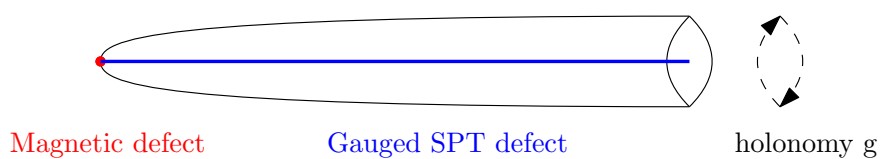

Magnetic defect      Gauged SPT defect      holonomy g

Figure 2: The magnetic defect in $G$ gauge theory with holonomy $g$ in the presence of topological action for $G$ becomes attached to a gauged SPT defect given by twisted compactification of the topological action on a circle with holonomy $g$.

$\omega_{D+1}$, and the gapped boundary has $K$ gauge group with action $\alpha_D$. The topological operators on the gapped boundary corresponds to the coset symmetry operators specified by $(G, K, \omega_{D+1}, \alpha_D)$.

As we have seen above, $\alpha_D$ can be shifted by $\alpha_D \to \alpha_D + \eta_D$ with an element $\eta_D \in Z^D(BG, U(1))$. When $[\eta_D] \in H^D(BG, U(1))$ is obtained by restricting the $G$ SPT response $[\eta'] \in H^D(BG, U(1))$ to $K \subset G$, the shift by $\eta_D$ represents the action of invertible global symmetry (called gauged SPT symmetry) of the bulk $G$ gauge theory on the boundary [38]. In this case, the shift $\alpha_D \to \alpha_D + \eta_D$ does not modify the symmetry category, since the gauged SPT defect $\eta_D$ induces the automorphism of the topological operators at the boundary. Meanwhile, when $\eta_D$ is not obtained by the restriction of $G$ SPT response to $K$, the shift $\alpha_D \to \alpha_D + \eta_D$ generally modifies the symmetry category of operators at the boundary, including its fusion rule. Nontrivial $\alpha_D$ that cannot be removed by $\eta_D$ represents a fractional response. The effect of $\alpha_D$ on the symmetry category structure will be discussed in detail in Sec. 2.3.

The bulk $G$ gauge theory has higher group symmetry that depends on the bulk cocycle $\omega_{D+1}$ as studied in [38]. In the rest of the section, we will use the bulk description to study properties of coset symmetry via bulk-boundary correspondence. We will shortly see that the higher-group structure in the bulk $G$ gauge theory affects the algebraic structure of the coset symmetry $(G, K, \omega_{D+1}, \alpha_D)$. Later in Sec. 3, we will use the bulk TQFT description to study anomalies of coset symmetries as obstructions to realization in symmetric, invertible phases.

## 2.2 Modified fusion rules from twist

The coset 0-form symmetry defects correspond to the magnetic defects in the bulk $G$ gauge theory. The pure magnetic defects are labeled by the conjugacy classes of $G$. When the gauge group is non-Abelian, the product of two conjugacy classes $[g_1], [g_2]$ give a direct sum of conjugacy classes $\oplus_g [g_1 g g_2 g^{-1}]$, and thus the fusion of magnetic fluxes is in general non-Abelian even in the absence of a topological twist $\omega_{D+1}$ for the $G$ gauge field.[3] When

---

[3]Note that in general, the fusion of pure magnetic fluxes can also include electric defects: these are the electric defects that become trivial when restricted to the common stabilizer subgroup of $G$ preserved by the fluxes.

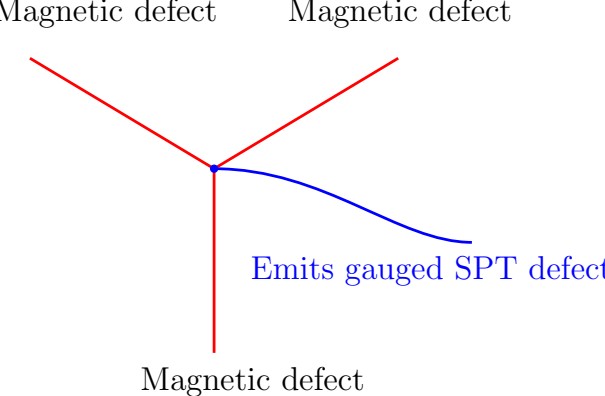

Figure 3: The topological action in $G$ gauge theory modifies the junction of magnetic or dyonic defects with additional gauged SPT defects.

the conjugacy class is in the center of $G$, the fusion of these fluxes is Abelian for $\omega_{D+1} = 0$.

As discussed in [38], the topological action $\omega_{D+1}$ of $G$ gauge theory can modify the fusion rule of the magnetic defects. This is because the magnetic defect of holonomy $g \in G$ extended in $(D-1)$ dimensions is attached to the additional gauged SPT defect in $D$ dimensions given by the slant product $i_g\omega_{D+1}$. The attached gauged SPT can be decomposed into two parts: one part, called $i_g^A\omega_{D+1}$ can make the magnetic defects obey non-invertible fusion rules. The other part, called, $i_g^B\omega_{D+1}$, modify the junction of magentic defects. They are explicitly expressed as follows.

$$i_g\omega_{D+1} = i_g^A\omega_{D+1} + \frac{1}{|\omega_{D+1}|}d\left(i_g^B\omega_{D+1}\right) \ , \tag{2.5}$$

where $|\omega_{D+1}|$ is the order of $[\omega_{D+1}]$ in $H^{D+1}(BG, U(1))$.

The above operations $i_g^A, i_g^B$ are defined as follows. Suppose that $[\omega_{D+1}]$ has the order of $k = |\omega_{D+1}|$ in $H^{D+1}(BG, U(1))$, then one can fix a cocycle representative which takes value in $2\pi\mathbb{Z}/k$ for any $(D+2)$ group elements:

$$\omega_{D+1} = \frac{2\pi}{k}\left(\omega_{D+1}\right)_k \ \ \text{mod}\ 2\pi\mathbb{Z} \ , \tag{2.6}$$

with a $\mathbb{Z}_k$ valued cocycle $(\omega_{D+1})_k$. Then, its slant product $i_g[(\omega_{D+1})_k]$ also has the value $2\pi\mathbb{Z}/k$. Now, suppose that the cohomology class $[i_g(\omega_{D+1})_k]$ has the order of $k'$ in $H^D(BG, U(1))$. Then there exists a cocycle representative $(i_g\omega_{D+1})'_{k'}$ which takes value in $2\pi\mathbb{Z}/k'$ for any $(D+1)$ group elements, and the cocycle $i_g[(\omega_{D+1})_k]$ is written as

$$i_g[(\omega_{D+1})_k] = \frac{k}{k'}\left(i_g\omega_{D+1}\right)'_{k'} + d(i'_g\omega_{D+1})_{k''}/k'' \ , \tag{2.7}$$

where the second term $d(i'_g\omega_{D+1})_{k''}/k''$ represents the difference from a cocycle representative $(i_g\omega_{D+1})'_{k'}$ by a coboundary. $(i'_g\omega_{D+1})_{k''}$ takes value in $2\pi\mathbb{Z}/k''$ with some integer $k''$. We

simply denote the above decomposition of $i_g\omega_{D+1}$ by

$$i_g\omega_{D+1} = i_g^A\omega_{D+1} + \frac{1}{|\omega_{D+1}|}d(i_g^B\omega_{D+1}) \ , \tag{2.8}$$

where $|\omega_{D+1}| = k$ is the order of $[\omega_{D+1}]$ in $H^{D+1}(BG, U(1))$, and $i_g^B\omega_{D+1} \in \frac{2\pi}{k''}\mathbb{Z}$.

The contribution from $i_g^A$ modifies the fusion rules of magnetic defects, while $i_g^B$ modifies the junction of magnetic defects [38] (see Figure 3). As a consequence, the fusion rule of the coset symmetry on the gapped boundary of the bulk $G$ gauge theory is also modified. The boundary condition breaks $G$ to subgroup $K$, and the boundary action for $K$ is $\alpha_D$. Below, we describe the fusion rule of coset symmetries realized by the topological defects of the boundary.

**Modified fusion from $i^A$** First, let us briefly recall how the slant product $i^A$ modifies the fusion rules of magnetic defects in the bulk $G$ gauge theory. The Dijkgraaf-Witten twist $\omega_{D+1}$ generally adds non-invertibility of the magnetic defects through the slant product $i^A$ mentioned above. For instance, the magnetic defects carrying holonomy in $Z(G)$ is invertible in untwisted gauge theory, but becomes non-invertible with the fusion rule [38]

$$V_g \times V_{g'} = V_{g+g'}\frac{1}{\mathcal{N}}\left(\sum_{\lambda \in Z(G)} \mathcal{W}_{i_\lambda i_g\omega_{D+1}}\right)\left(\sum_{\lambda' \in Z(G)} \mathcal{W}_{i_{\lambda'} i_{g'}\omega_{D+1}}\right)\Big/\{\mathcal{W}_{i_{\lambda''} i_{g+g'}(\omega_{D+1})} : \lambda'' \in Z(G)\} \ ,$$

$$\tag{2.9}$$

and

$$V_g \times \mathcal{W}_{i_\lambda i_g\omega_{D+1}} = V_g \ , \quad \forall \lambda \in Z(G) \ , \tag{2.10}$$

where $V_g$ is the magnetic defect carrying the center holonomy $g \in Z(G)$, and $\mathcal{W}_\omega$ is the gauged SPT defect labeled by the group cohomology $\omega$.

Reflecting such fusion rules of the magnetic defects in the bulk, the fusion rules of coset symmetry on the boundary are modified if $\omega_{D+1}$ satisfies $\omega_{D+1}|_K = d\alpha$ and also $[i_g^A\omega_{D+1}]_K \neq 0$. In other words, although the restriction of $\omega_{D+1}$ to subgroup $K$ is exact, it is no longer exact if we first take the slant product of $\omega_{D+1}$ with respect to $G$.

An example is $G = \mathbb{Z}_2 \times G'$, with $\omega_{D+1} = a \cup \omega'_D$ where $a$ is the $\mathbb{Z}_2$ 1-cocycle and $\omega'$ is a $G'$ cocycle of degree $D$. The subgroup is $K = G'$. The cocycle $\omega_{D+1}$ restricted to $K$ is trivial, since it vanishes for $a = 0$. However, if we first take the slant product with respect to the nontrivial holonomy in $\mathbb{Z}_2$, this gives $i_g\omega_{D+1} = \omega'_D$ which can be a nontrivial cocycle of $K = G'$.

The property that the coset symmetry is attached to a gauged SPT defect implies that the coset symmetry becomes "more non-invertible" in the presence of twists: for magnetic defects with center holonomy, there are non-invertible fusion rule obtained by replacing $i_\lambda i_g\omega_{D+1}$ in (2.9), (2.10) to $[i_\lambda i_g\omega_{D+1}]_K$.

**Modified fusion from $i^B$** Let us briefly recall how the slant product $i^B$ modifies the fusion rule of magnetic defects in the bulk $G$ gauge theory. The slant product $i^B$ modifies the $(D-2)$-dimensional junction of the magnetic defects by attaching the end of the gauged SPT defects in $(D-1)$ dimensions. In other words, the fusion algebra of the magnetic defects is extended by the gauged SPT defects.

For instance, let us consider the 1-form symmetry generated by the magnetic defects $V_g$ of center holonomy $g \in Z(G)$. Since the twist makes some of these magnetic defects non-invertible, the invertible magnetic defects are labeled by

$$Z_\omega(G) \equiv \{g \in Z(G): \quad [i_g^A \omega_{D+1}] = 0 \in H^D(BG, U(1))\} \ . \tag{2.11}$$

The 1-form symmetry of these magnetic defects then becomes the central extension

$$1 \to H^{D-1}(BG, U(1)) \to \mathcal{G}^{(1)} \to Z_\omega(G) \to 1 \ , \tag{2.12}$$

which is characterized by the second cohomology $\Omega_\omega$ [38]

$$\Omega_\omega(g, g') = \frac{1}{|\omega_{D+1}|} \left( i_g^B \omega_{D+1} + i_{g'}^B \omega_{D+1} - i_{g+g'}^B \omega_{D+1} \right) \ , \tag{2.13}$$

where $|\omega_{D+1}|$ is the order of $[\omega_{D+1}]$ in the group $H^{D+1}(BG, U(1))$.

Let us now discuss how the above central extension affects the structure of the coset symmetry. We define $Z_\omega(G/K) := Z_\omega(G)/(Z_\omega(G) \cap K)$, which is the group of nontrivial coset symmetry defects on the boundary carrying the center holonomy. Here let us consider the cases where $Z_\omega(G)$ has the form of direct product, $Z_\omega(G) = Z_\omega(G/K) \times K'$, with $K' = Z_\omega(G) \cap K$. The generic cases will be studied in Sec. 2.3. In this case, $\Omega_\omega$ directly induces the central extension of the coset symmetry, and defines an extension class in $H^2(Z_\omega(G/K), H^{D-1}(BK, U(1)))$. This is obtained by restriction to the group $Z_\omega(G/K) \subset Z_\omega(G)$, together with the restriction of gauge group $G \to K$ in $H^{D-1}(BG, U(1))$.

The coset 0-form symmetry at the boundary then has the following central extension due to the bulk cocycle $\omega_{D+1}$,

$$1 \to H^{D-1}(BK, U(1)) \to \mathcal{G}^{(0)}_{G/K} \to Z_\omega(G/K) \to 1 \ , \tag{2.14}$$

characterized by the restriction of $\Omega_\omega$. As we will see in Sec. 2.3, in the presence of the boundary twist $\alpha_D$ the class for the central extension $\Omega_\omega$ will be further modified by slant product of $\alpha_D$.

In the following, we will give concrete examples to illustrate how the cocycles modify the fusion rules.

### 2.2.1 Example of modified fusion rules from $i^A$

A well-known example for the modified fusion rule of coset symmetry is the twisted $\mathbb{Z}_2^3$ gauge theory in $(2+1)$D, which is equivalent to the $D_8$ gauge theory and its boundary realizes the non-invertible twisted coset symmetry $\mathrm{Rep}(D_8)$.

Let us consider $G = \mathbb{Z}_2^3$ and $K = \mathbb{Z}_2^2$ in $D = 2$ spacetime dimension, and $\omega_{D+1} = (-1)^{a_1 \cup a_2 \cup a_3}$ where $a_1, a_2, a_3$ generate the $\mathbb{Z}_2$ 1-cocycles for the three $\mathbb{Z}_2$ symmetries. The subgroup $K$ corresponds to the boundary condition $a_3| = 0$.

In the bulk $G$ gauge theory, the electric charges $e_i$ i.e. Wilson lines of $a_i$ obey $\mathbb{Z}_2^3$ fusion rule, while the magnetic defects $m_1, m_2, m_3$ obey the fusion rule

$$m_i \times m_i = 1 + e_j + e_k + e_j e_k, \quad i, j, k \text{ distinct}. \tag{2.15}$$

The electric charge $e_3$ are condensed on the boundary, while $e_1, e_2$ are not. The magnetic defect $m_3$ can move parallel to the boundary and gives the coset symmetry, it obeys the fusion rule

$$m_3 \times m_3 = 1 + e_1 + e_2 + e_1 e_2, \tag{2.16}$$

which corresponds to the non-invertible fusion rule of $\text{Rep}(D_8)$. The magnetic defects $m_1, m_2$ terminates on the boundary, and their fusion rules are $m_1 \times m_1 = 2(1+e_2)$, $m_2 \times m_2 = 2(1+e_1)$, where we have used $e_3 \sim 1$.

Such coset symmetry fusion rule is different from the "untwisted" coset symmetry with $\omega_{D+1} = 0$, where the coset symmetry obeys $\mathbb{Z}_2$ fusion rule up to tensor product with $\text{Rep}(\mathbb{Z}_2^2)$.

### 2.2.2  Example of modified fusion rules from $i^B$

Let us consider another example to illustrate the modified fusion rule from the $i^B$ term. Consider $D = 4$, and $G = \mathbb{Z}_2 \times \mathbb{Z}_2$, with

$$\omega_5 = (-1)^{a_1^4 \cup a_2} = (-1)^{(da_1/2) \cup (da_1/2) \cup a_2}, \tag{2.17}$$

where $a_1, a_2$ generate the $\mathbb{Z}_2$ one-cocycles of the two $\mathbb{Z}_2$'s, and the last expression we take a lift of $a_1$ to $\mathbb{Z}_4$ cochain. The subgroup is the first $\mathbb{Z}_2$, $K = \mathbb{Z}_2$. The coset symmetry on the boundary corresponds to the magnetic defect of $a_2$.

In the 5D bulk gauge theory, the cocycle $\omega_5$ modifies the property of the magnetic defect for $a_2$, which is created by volume operator supported on some volume $V$. The twisted compactification of $\omega_5$ on a circle with $a_2$ holonomy is a total derivative,

$$\int_{M_4} (da_1/2) \cup (da_1/2) = \int_V a_1 \cup da_1/4, \tag{2.18}$$

where $\partial M_4 = V$. Thus the magnetic volume operator carries "fractional" gauged SPT defect described by the right hand side above. This implies that the junction of the magnetic defect emits the gauged SPT defect

$$(-1)^{\int a_1 \cup da_1/2} = (-1)^{\int a_1^3}, \tag{2.19}$$

which represents the response of the $\mathbb{Z}_2$ Levin-Gu SPT in (2+1)D. In other words, the 1-form symmetry in the bulk generated by the magnetic defect for $a_2$ becomes the central extension

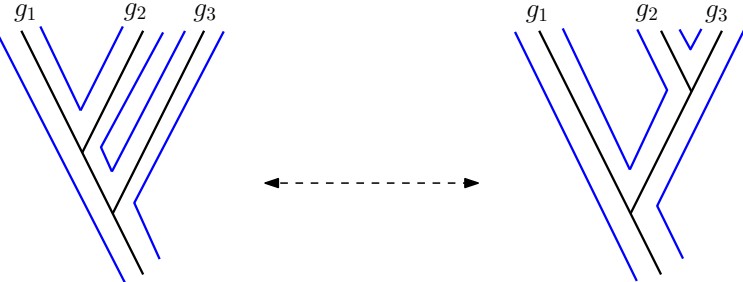

Figure 4: Associator of twisted coset symmetry $(G, K, \omega_{D+1}, \alpha_D)$ is inherited from the associator $\omega_{D+1}$ of twisted $G$ symmetry using the sandwich construction. The blue lines are interfaces of Dirichlet boundary condition of the $K$ gauge field, and the black lines are the $G$ symmetry defects.

$\mathbb{Z}_2 \to \mathbb{Z}_4 \to \mathbb{Z}_2$ extended by the gauged SPT defect. Let us denote the excitation by $s_{\mathrm{LG}}$ stands for the Levin-Gu SPT, then the fusion rule is

$$m_2 \times m_2 = s_{\mathrm{LG}} \ . \tag{2.20}$$

On the boundary, the SPT defect for $a_1$ is nontrivial. and thus the fusion rule of the coset symmetry on the boundary is modified by the presence of nontrivial cocycle $\omega_5$.

## 2.3 Effect of $\alpha_D$ on coset symmetry

Different choices of $\alpha_D$ are related by $\alpha'_D = \alpha_D + \eta_D$ with $\eta_D \in H^D(BK, U(1))$ the $K$ SPT response. When $\eta_D$ is obtained by restricting $G$ SPT response to the subgroup $K \subset G$, this correspond to fusing the boundary with a gauged SPT defect for $G$ in the bulk. In that case, the shift $\alpha_D \to \alpha_D + \eta_D$ does not modify the symmetry category, as the gauged SPT defect induces the automorphism of the topological operators at the boundary. However, note that the shift by $\eta_D$ in general does not have such bulk interpretation. When $\eta_D$ is not obtained by the bulk gauged SPT defect, the shift $\alpha_D \to \alpha_D + \eta_D$ can modify the symmetry category.

The boundary action $\alpha_D$ means that when a magnetic defect for $K$ terminates on the boundary to give rise to dynamical $K$ gauge field holonomies on the boundary, the ending locus has additional gauged SPT defects for the $K$ remnant gauge group as discussed in [38]. Such changes of condensed objects by shifting $\alpha \to \alpha_D + \eta_D$ modify the symmetry category.

Concretely, let us consider the group of center magnetic defects in the bulk $Z_\omega(G)$, as defined in (2.11). The bulk center symmetry $Z_\omega(G)$ is given by the central extension of the coset symmetry $Z_\omega(G/K)$ at the boundary,

$$Z_\omega(G) \cap K \to Z_\omega(G) \to Z_\omega(G/K) \ . \tag{2.21}$$

Let us write the above extension class $\nu \in H^2(Z_\omega(G/K), Z_\omega(G) \cap K)$. While in Sec. 2.2.2 we have seen that the coset symmetry $Z_\omega(G/K)$ gets extended due to the bulk cocycle $\omega_{D+1}$

through the slant product $i^B$, the boundary cochain $\alpha_D$ also extends the coset symmetry through its slant product. This is because the gauged $K$ SPT defects generally get identified with the $K$ magnetic defects at boundary up to condensed objects, meaning that the gauged $K$ SPT defects replace the group $Z_\omega(G) \cap K$ in (2.21) in the structure of coset symmetry.

The coset symmetry has the form of the central extension with the gauged $K$ SPT defect,

$$1 \to H^{D-1}(BK, U(1)) \to \mathcal{G}^{(0)}_{G/K} \to Z_\omega(G/K) \to 1 \;, \tag{2.22}$$

whose extension class is characterized by

$$\Omega(\tilde{g}, \tilde{g}') = \frac{1}{|\omega_{D+1}|} \left( i^B_{(\tilde{g},0)}\omega_{D+1} + i^B_{(\tilde{g}',0)}\omega_{D+1} - i^B_{(\tilde{g}+\tilde{g}',\nu(\tilde{g},\tilde{g}'))}\omega_{D+1} \right) |_K + i_{\nu(\tilde{g},\tilde{g}')}\alpha_D \;, \tag{2.23}$$

where $\tilde{g}, \tilde{g}' \in Z_\omega(G/K)$, and we label the elements of $Z_\omega(G)$ by a pair $(\tilde{g}, \tilde{k}) \in Z_\omega(G/K) \times (Z_\omega(G) \cap K)$. Also $|_K$ means the restriction to the group $K$. Below, we will provide an example where $\alpha_D$ modifies the fusion rule by the extension.

### 2.3.1 Example of modified fusion rules from $\alpha_D$

Let us consider the untwisted coset symmetry in $D = 2$ with

$$(G, K, \omega_3, \alpha_2) = (\mathbb{Z}_4 \times \mathbb{Z}_2, \mathbb{Z}_2 \times \mathbb{Z}_2, 0, 0). \tag{2.24}$$

This corresponds to the gapped boundary of $\mathbb{Z}_4 \times \mathbb{Z}_2$ gauge theory in (2+1)D. Let us denote the electric and magnetic particles of $\mathbb{Z}_4, \mathbb{Z}_2$ gauge theory as $\{e, m\}, \{e', m'\}$ respectively. The condensed particles at the boundary is then generated by $m^2, e^2, m'$. The symmetry at the boundary is generated by the lines of $m, e, e'$. They generate the anomalous $\mathbb{Z}_2^3$ symmetry, where the first two $\mathbb{Z}_2$ symmetries for $m, e$ have the mixed 't Hooft anomaly. This mixed 't Hooft anomaly is understood from $i$ mutual braiding between $\mathbb{Z}_4$ particles $e$ and $m$, so that one $\mathbb{Z}_2$ symmetry defect carries the fractional charge $1/2$ under the other $\mathbb{Z}_2$ symmetry.

Then let us turn on the boundary action $\alpha_2 = (-1)^{a \cup a'}$, where $a, a'$ denote two $\mathbb{Z}_2$ gauge fields at the boundary:

$$(G, K, \omega_3, \alpha_2) = (\mathbb{Z}_4 \times \mathbb{Z}_2, \mathbb{Z}_2 \times \mathbb{Z}_2, 0, (-1)^{a \cup a'}). \tag{2.25}$$

The SPT action $\alpha_2$ shifts the condensed particles; it exchanges the particles by $m^2 \to m^2 e', m' \to m'e$ at the boundary, so the condensed particles are now generated by $m^2 e', e^2, m'e$. Note that this action of $\alpha_2$ on condensed anyons is not an automorphism of the bulk $\mathbb{Z}_4 \times \mathbb{Z}_2$ gauge theory, and $\alpha_2$ cannot be obtained from the gauged SPT defect in the bulk. The symmetry at the boundary is still generated by the lines of $m, e, e'$, but they now generate the non-anomalous $\mathbb{Z}_4 \times \mathbb{Z}_2$ symmetry, as $m^2 \sim e'$ up to condensed particles. We note that the electric particle $e'$ gives the central extension of the coset symmetry $m$ from $\mathbb{Z}_2$ to $\mathbb{Z}_4$; $e'$ corresponds to the slant product of $\alpha_2$ shown in the description of (2.23). This shows that different choices of $\alpha_2$ can modify the fusion rule of the coset symmetry.

## 2.4 Modified Frobenius-Schur indicator from twist

As discussed in [39], the Frobenius-Schur indicator $\kappa_a$ is a $\mathbb{Z}_2$ valued piece of data associated with self-dual topological lines $a = \bar{a}$. It is part of the $F$ symbols for such topological lines, in particular $[F_a^{a\bar{a}a}]_{11} = [F_a^{aaa}]_{11}$ where 1 is the vacuum object. There is also a Frobenius-Schur indicator as part of the associator data for 0-form symmetries in 1+1d. These two Frobenius-Schur indicators are related to each other: the topological defects whose fusion are modified by the Frobenius-Schur indicator in 1+1d can be thought of as living on the boundary of a $\mathbb{Z}_2$ 0-form symmetry domain wall that extends to 2+1d bulk, and the Frobenius-Schur indicator is given by whether the $\mathbb{Z}_2$ symmetry has nontrivial bosonic SPT (see e.g. [6, 7]). After gauging the $\mathbb{Z}_2$ symmetry in the bulk, the $\mathbb{Z}_2$ symmetry fluxes give rise to new self-dual topological lines, whose associators carry information about the bosonic $\mathbb{Z}_2$ SPT. Such SPTs can be described a 3-cocycle for group $\mathbb{Z}_2$ [40]. There is only one nontrivial such cocycle, corresponding to the Levin-Gu phase [41], and this is consistent with the Frobenius-Schur indicator being $\mathbb{Z}_2$ valued in 2+1d.

In analogy, in this work, we define a generalized Frobenius-Schur indicator as a particular part of the data describing the associator of topological defects in 3+1d, in 1-1 correspondence to SPTs of the bulk symTFT. For bulk symmetry $G$, the generalized Frobenius-Schur indicator is labeled by $G$-SPT in the bulk, as described by a $(D+1)$-cocycle $\omega_{D+1}$ for group $G$ [40], and they are phase factors that relate different ways of fusing $(D+1)$ symmetry defects for $G$. Such a cocycle corresponds to an anomaly for the $G$ symmetry (both an obstruction to realization in SPT phases and obstruction to gauging the $G$ symmetry). The bulk TQFT description for the Frobenius-Schur indicator is the Boltzmann weight given by the phase factor associated with intersection of $(D+1)$ symmetry defects at a point.

The generalized Frobenius-Schur indicator for twisted coset symmetry can be derived using the "sandwich construction" for the coset symmetry defects. Since the $G$ symmetry defects in the middle have associator given by $\omega_{D+1}$, it also gives the associator of the twisted coset symmetry defects. See Figure 4 for the illustration of the associator in $D = 2$.

For example, let us start with a system with anomalous $G$ symmetry in $D$ spacetime dimensions with anomaly given by cocycle $\omega_{D+1}$, such that the subgroup $K$ is non-anomalous: $\omega_{D+1}|_K = d\alpha_D$. Then we can gauge the $K$ subgroup symmetry, and the new system has twisted coset symmetry with associator again given by $\omega_{D+1}$.

## 2.5 Redundancy in definition

The presentation of twisted coset symmetry by $(G, K, \omega_{D+1}, \alpha_D)$ is in general redundant: there can be two quadruples $(G, K, \omega_{D+1}, \alpha_D)$ and $(G', K', \omega'_{D+1}, \alpha'_D)$ that describe the same coset symmetry. Here let us describe several occasions where such redundancy is observed.

- For instance, this can happen when two different labels of the bulk gauge theory $(G, \omega_{D+1})$ results in the same TQFT. Such redundancy in labeling of the twisted gauge

theory is ubiquitous in (2+1)D with $D = 2$. For instance, the twisted $\mathbb{Z}_2^3$ gauge theory in (2+1)D is equivalent to $D_8$ gauge theory. Accordingly, the coset symmetry $(\mathbb{Z}_2^3, \mathbb{Z}_2^2, \omega_3, 0)$ is equivalent to $(D_8, D_8, 0, 0)$ when $\omega_3 = (-1)^{a_1 a_2 a_3}$ with $\mathbb{Z}_2^3$ gauge fields, as described in Sec. 2.2.1.

- This can happen even when the labels of the bulk gauge theory are identical. For instance, some of (1+1)D SPT phases with finite $G$ symmetry have a trivial torus partition function, but become nontrivial on closed surfaces with higher genus. Let us take such a $G$ SPT phase given by $\alpha \in H^2(BG, U(1))$. Then, the coset symmetries $(G, G, 0, \alpha)$ and $(G, G, 0, 0)$ in $D = 2$ are identical, both of which are given by Rep$(G)$. This corresponds to a pair of distinct gapped boundaries of $G$ gauge theory whose condensed particles (Lagrangian algebra anyon) are identical, though the algebraic structure of the Lagrangian algebra is still distinguished by their multiplication morphisms. The gapped boundary $(G, G, 0, \alpha)$ is obtained by acting the 0-form symmetry of $G$ gauge theory generated by the gauged SPT defect $\alpha$ on the boundary $(G, G, 0, 0)$. This 0-form symmetry does not permute anyons, and only acts on the junction of anyons, dubbed a soft symmetry [42, 43].

- This can happen even when the bulk gauge groups are distinct. For instance, it is known that there exists a pair of distinct finite groups $G_1, G_2$ satisfying Rep$(G_1) =$ Rep$(G_2)$ as a fusion category [44] (without symmetric structure). Therefore, in $D = 2$ the coset symmetry $(G_1, G_1, 0, 0)$ is identical to $(G_2, G_2, 0, 0)$ where both are given by Rep$(G_1)$.

## 2.6 Examples of theories with twisted coset symmetries

We now give a few examples of systems with twisted coset symmetries, both on the lattice and in the continuum.

### 2.6.1 Lattice models with twisted coset symmetry

To be concrete, we will construct a lattice model with twisted coset non-invertible symmetry. We will begin by constructing a lattice model with anomalous invertible 0-form symmetry, and then gauge a non-anomalous and non-normal subgroup.

Let us start with 2+1D toric code enriched with $S_3 \times \mathbb{Z}_2$ 0-form symmetry using the anomalous $\mathbb{Z}_2 \times \mathbb{Z}_2$ one-form symmetry generated by the electric and magnetic line operators. In terms of the one-form symmetry background fields $B^e, B^m$, the symmetry enrichment is given by the relation [45]

$$B^e = A^* \eta_2, \quad B^m = A^* \eta_1 \cup A' \quad \text{mod } 2 , \tag{2.26}$$

where $\eta_2$ is the nontrivial generator in $H^2(BS_3, \mathbb{Z}_2) = \mathbb{Z}_2$ and $\eta_1$ is the nontrivial generator in $H^1(BS_3, \mathbb{Z}_2) = \mathbb{Z}_2$ (i.e. the charge conjugation gauge field), $A$ is the background gauge

field for $S_3$ and $A'$ is the background gauge field for $\mathbb{Z}_2$. Due to the mixed anomaly between the electric and magnetic one-form symmetry, the $S_3 \times \mathbb{Z}_2$-enriched toric code has anomaly described by the bulk term

$$\pi \int B^e \cup B^m = \pi \int A^*(\eta_1 \cup \eta_2) \cup A' \ . \tag{2.27}$$

The lattice model with such symmetry enrichment can be constructed following the method in [46].

Next, we gauge the non-normal $\mathbb{Z}_2$ subgroup symmetry in $S_3$. Since the anomaly vanishes for $A' = 0$, i.e., absence of gauge field for the other factorized $\mathbb{Z}_2$, we are allowed to gauge this symmetry. The resulting theory has $(S_3/\mathbb{Z}_2) \times \mathbb{Z}_2$ twisted coset non-invertible symmetry:

$$(G = S_3 \times \mathbb{Z}_2, \ K = \mathbb{Z}_2 \times 1, \ \omega_4 = \eta_1 \cup \eta_2 \cup x_1, \ \alpha_3 = 0) \ , \tag{2.28}$$

where $x_1$ is the generator of $H^1(B\mathbb{Z}_2, U(1)) = \mathbb{Z}_2$.

### 2.6.2 Critical points with twisted coset symmetry

We will present a critical theory that has fractional response for coset non-invertible symmetry. The example is given by massless Dirac fermion in 2+1d, coupled to $\mathbb{Z}_2$ gauge field where the $\mathbb{Z}_2$ gauge transformation complex conjugates the Dirac fermion. Before gauging the $\mathbb{Z}_2$ symmetry, the theory has $O(2)$ symmetry, and the $O(2)$ symmetry has fractional response

$$O(2)_{1/2,1/2} + \text{Dirac fermion} \ , \tag{2.29}$$

where the subscript denotes fractional Chern-Simons level for both the continuous and the $\mathbb{Z}_2$ part, following the notation in [35]. Since the $\mathbb{Z}_2$ symmetry complex conjugates the Dirac fermion that consists of two Majorana fermions, it leaves one of the Majorana fermion invariant and flips the sign of the other Majorana fermion. After gauging the $\mathbb{Z}_2$ symmetry, the symmetry is the coset symmetry $O(2)/\mathbb{Z}_2$. The coset symmetry has $\alpha = \frac{1}{2}\eta$ where $\eta$ is the effective action for the minimal $\mathbb{Z}_2$ Chern-Simons term given by gauging the $\mathbb{Z}_2$ symmetry in the root $\mathbb{Z}_2$ fermionic SPT phase. The coset symmetry has $\omega$ given by an exact cocycle, which contains fractional response of $O(2)/\mathbb{Z}_2$ symmetry.

We can also consider discrete symmetry. For example, take a Dirac fermion, and focus on the $\mathbb{D}_{4n} = \mathbb{Z}_{2n} \rtimes \mathbb{Z}_2 \subset O(2) = U(1) \rtimes \mathbb{Z}_2$ subgroup of the previous example. The previous discussion carries over to give massless fermion with fractional response for the coset $(\mathbb{Z}_{2n} \rtimes \mathbb{Z}_2)/\mathbb{Z}_2$ symmetry.

## 3 Anomalies as Absence of SPT Phases

An important application of anomalies is their constraints on the dynamics. In this section we will study whether the coset symmetry can be realized in a trivially gapped phase. If not,

then we will say the coset symmetry has intrinsic anomalies as obstructions to SPT realization. Anomalies of the coset symmetries generally leads to the tight dynamical constraints such as symmetry-enforced gaplessness, as we will see below.

## 3.1    Anomaly conditions

While the computation of the properties such as fusion rules and associators of the coset symmetry from the data $(G, K, \omega_{D+1}, \alpha_D)$ can be difficult, we can use the bulk-boundary correspondence to understand the intrinsic anomalies of the coset symmetry, i.e. the anomalies present in any systems with the coset symmetry.

The bulk-boundary correspondence has been widely used in studying the anomaly, i.e. obstruction to SPT realization, of non-invertible symmetries [6, 7]. Here, the bulk is described a twisted $G$ gauge theory in $(D + 1)$ spacetime dimensions with cocycle $\omega_{D+1}$, and the coset symmetry is on the boundary with the boundary condition that breaks $G$ to subgroup $K$, with $\omega_{D+1}|_K = d\alpha_D$. The topological operators on this gapped boundary with the boundary action $\alpha_D$ describes the twisted coset symmetry $(G, K, \omega_{D+1}, \alpha_D)$. A generic gapped system with this coset symmetry can be given by the bulk TQFT on a thin interval, where one end of the interval is given by the gapped boundary $G \to K$ with the boundary topological action $\alpha_D$, while the other end is given by another choice of gapped boundary.

The question we ask is the following: can we realize an SPT phase with the coset symmetry from the bulk-boundary description? If there is no such an SPT phase, then the symmetry has anomaly as obstruction to such realization.

To engineer an SPT phase, we need to choose a gapped boundary of the bulk $G$ gauge theory with topological action $\omega_{D+1}$, such that the bulk on the interval with the gapped boundary on the other end reduces to an SPT phase with the coset symmetry. This requires that no operators can simultaneously end on the two boundaries. The operators in the bulk $G$ gauge theory can be generated by the basic Wilson line and magnetic operators under fusion or condensation, and we will only need to see if the these basic operators can simultaneously end on the two boundaries.

The gapped boundaries for the $G$ gauge theories are labeled by

- A subgroup $K' \subset G$ such that $[\omega_{D+1}|_{K'}] = 0$.

- A choice of boundary topological term for $K'$, i.e. choice of $\alpha'_D$ in $\omega_{D+1}|_{K'} = d\alpha'_D$.

For the reduction of the bulk on the interval with the above gapped boundary on other end to give rise to an SPT phase, we need to require the following conditions:

(1) The subgroups $K, K'$ for the gapped boundaries need to satisfy $[\omega_{D+1}|_K] = 0, [\omega_{D+1}|_{K'}] = 0$, i.e., they become exact cocycles under pullback using the inclusion map $K, K' \subset G$. This also guarantees that the magnetic operators of $K, K'$ holonomy are not attached to gauged SPT operators [38].

(2) There are no magnetic operators that connect the two boundaries, $K \cap K' = 1$.

(3) There are no Wilson line operators that connect the two boundaries: the only irrep $R$ whose decomposition under $K$ and $K'$ both contain identity is the trivial irrep $R = 1$.

If all possible choices of the subgroup $K' \subset G$ violate at least one of the above conditions (1)-(3), then the coset symmetry has an intrinsic anomaly – it cannot be realized by SPT phases, and any systems with such coset symmetry cannot flow to a trivially gapped phase.

We note that the difference in the data $\alpha_D$ does not affect the SPT realization – since it is purely on the boundary, it does not contribute to this anomaly.

## 3.2  SPT phases with twisted coset symmetry $(H \bowtie K)/K$

Let us first study the coset symmetry without anomalies, where one can explicitly find the SPT phases. We show that the twisted coset symmetry $(G, K, \omega_{D+1}, \alpha_D)$ can be realized in the SPT phase if the following two conditions are satisfied:

- $G$ is expressed as the bicrossed product with some subgroup $H \subset G$ as $G = H \bowtie K$. Namely, for two subgroups $H, K$ of $G$, $G$ can be expressed as $G = HK$ and $H \cap K = \{\mathrm{id}\}$. Equivalently, each group element $g \in G$ has a unique expression as $g = hk$ with $h \in H, k \in K$.[4]

- There exists a choice of the above $H \subset G$ satisfying $[\omega_{D+1}|_H] = 0$.

To check this, let us consider a bulk symmetry TQFT in $(D+1)$D on an interval, where the $G$ gauge theory with the twist $\omega_{D+1}$ is sandwiched by a pair of gapped boundary conditions. One of the gapped boundary breaks the gauge symmetry $G \to K$, while the other breaks as $G \to H$. We note that this SSB pattern $G \to H$ is made possible thanks to the above two conditions. The coset symmetry defects are realized as the topological operators at the boundary $G \to K$. One can see that this system realizes the SPT after shrinking the interval by the following:

- Since $H \cap K = \{\mathrm{id}\}$, there are no magnetic operators ending on both boundaries.

- One can also check that there are no irrep of $G$ such that its decomposition under the subgroups $H, K$ simultaneously contains the trivial representations of $H, K$. This is shown in Appendix A. It implies that there are no Wilson line stretching between two boundaries.

Since the gapped boundary $G \to H$ can be additionally twisted by a group cohomology $[\eta_D] \in H^D(BH, U(1))$, a subclass of SPT with $(H \bowtie K)/K$ symmetry in $D$ spacetime dimensions is classified by the group cohomology $H^D(BH, U(1))$. For untwisted coset symmetries with $\omega_{D+1} = 0$, the SPT can be obtained by starting with the SSB phase with $G \to H$ with the SPT action $\eta_D$ for the unbroken symmetry, and then gauging the broken $K$

---

[4]When $H$ is a normal subgroup, the bicrossed product reduces to the semidirect product $H \rtimes K$.

symmetry. An explicit lattice model for an SPT with untwisted $(H \bowtie K)/K$ symmetry will be presented in Sec. 4.1.

## 3.3 Anomalies of twisted coset symmetry: anomaly from bulk cocycle

The twist $\omega_{D+1}$ in the coset symmetry typically leads to the nontrivial anomalies. To illustrate the obstruction to SPT phase with coset symmetry due to nontrivial twist $\omega_{D+1}$, let us consider the example $G = G_0 \times G_1$ and $K$ is given by a non-normal subgroup of $G_0$. Moreover, there is nontrivial $\omega_{D+1}$ given by topological term for $G_1$. Then on any gapped boundary, $G_1$ needs to break to subgroup $K_1$ such that $\omega_{D+1}|_{K_1} = d\alpha$, and in particular the reference boundary $G \to K$ breaks $G_1$ completely. This means that there are Wilson lines of $G_1$ that can connect the two boundaries and thus violating condition (3) presented in Sec. 3.1. This gives an example of anomalous twisted coset non-invertible symmetry.

Below we will also discuss examples of coset non-invertible symmetries with trivial bulk cocycle that cannot be realized in SPT phases.

## 3.4 Anomalies of untwisted coset non-invertible symmetry

In Sec. 3.2 we have seen that the untwisted $(H \bowtie K)/K$ symmetry is non-anomalous and admits realization in SPT phases. Meanwhile, if the untwisted coset symmetries $G/K$ does not have an expression $G = H \bowtie K$ with some subgroup $H \subset G$, one can show that $G/K$ is anomalous. We will demonstrate this statement in general in Sec. 3.6.1. Here, let us provide an example of such anomalous finite coset non-invertible symmetry without twist.

### 3.4.1 $A_5/\mathbb{Z}_2$ symmetry

Consider an alternating group $A_5$ and its subgroup $K = \mathbb{Z}_2$. We will focus on the case with $D = 3$ for simplicity. Since $A_5$ is simple, $K$ is not a normal subgroup and $A_5/\mathbb{Z}_2$ is non-invertible. This is a minimal coset symmetry. Also, $A_5$ does not have a subgroup $H$ with order $30 = |A_5|/|\mathbb{Z}_2|$, so $A_5$ cannot be expressed as the bicrossed product $H \bowtie K$ with $K = \mathbb{Z}_2$. So $A_5/\mathbb{Z}_2$ is a candidate of the coset non-invertible symmetry that does not admit realization in SPT phases.

Let us consider the symmetry TQFT for the coset $G/K$ symmetry, which is the $G = A_5$ gauge theory. One boundary breaks the gauge group $G \to K$, and the other breaks $G \to K'$ with some subgroup $K' \subset G$.

The (2+1)D $A_5$ gauge theory has the anyons labeled by $([g], \pi)$ with $[g] \in \mathrm{Conj}(G)$ is the conjugacy class of $g \in G$, and $\pi$ is an irrep of the centralizer $Z(g)$ of $g \in G$. The anyons with $g = 1$ corresponds to the electric particles carrying the irrep of $G$.

There are five irreps of $A_5$ including the trivial one. The character table is given as follows:

$$
\begin{pmatrix}
1 & 1 & 1 & 1 & 1 \\
4 & 0 & 1 & -1 & -1 \\
5 & 1 & -1 & 0 & 0 \\
3 & -1 & 0 & \frac{1+\sqrt{5}}{2} & \frac{1-\sqrt{5}}{2} \\
3 & -1 & 0 & \frac{1-\sqrt{5}}{2} & \frac{1+\sqrt{5}}{2}
\end{pmatrix} ,
\tag{3.1}
$$

where each row corresponds to the irrep $\pi_1, \pi_4, \pi_5, \pi_3, \pi_3'$.

The Lagrangian algebra for the boundary condition $A_5 \to \mathbb{Z}_2$ is given as follows:

$$
\mathcal{A}_K = (1,1) \oplus 2(1,\pi_4) \oplus 3(1,\pi_5) \oplus (1,\pi_3) \oplus (1,\pi_3') \oplus ([(1,2)(3,4)],1) \oplus ([(1,2)(3,4)],\sigma) ,
\tag{3.2}
$$

where $\sigma$ is the irrep of $Z((1,2)(3,4)) = \mathbb{Z}_2^2$ satisfying $\sigma((1,2)(3,4)) = 1, \sigma((1,3)(2,4)) = -1$.

Note that the boundary condition $G \to K$ can condense all electric particles of the theory, in the sense that $\mathcal{A}_K$ contains all electric particles in the sum. Nevertheless, this boundary condition is different from the Dirichlet boundary condition $G \to 1$ with the Lagrangian algebra

$$
\mathcal{A}_1 = (1,1) \oplus 4(1,\pi_4) \oplus 5(1,\pi_5) \oplus 3(1,\pi_3) \oplus 3(1,\pi_3') .
\tag{3.3}
$$

The above form of $\mathcal{A}_K$ immediately tells that the coset symmetry $A_5/\mathbb{Z}_2$ is anomalous. To see this, we check that the condensed particles for any boundary condition $G \to K'$ has a nontrivial overlap with $\mathcal{A}_K$. When $K' \neq G$, some nontrivial electric particle $(1,\pi_j)$ is condensed at the $G \to K'$ boundary condition, which leads to an overlap.[5] When $K' = G$, $([(1,2)(3,4)],1)$ is condensed, which again leads to an overlap. This completes the argument that $A_5/\mathbb{Z}_2$ is anomalous. In Appendix B, we introduce another example of untwisted coset symmetry which is anomalous, given by $A_6/A_5$.

## 3.5 Dynamical scenario: symmetric gapped phases

In Sec. 3.2 we have seen that the coset symmetry $G/K$ with the twist $\omega_{D+1}$ admits realization in SPT phases, if $G$ is expressed as a bicrossed product $G = H \bowtie K$ with a subgroup $H \subset G$ such that $[\omega_{D+1}|_H] = 0$. In this section, we will show that once a finite group $G$ is expressed

---

[5]In general, an irrep $\pi$ is condensed at the $G \to K'$ boundary iff $\sum_{k' \in K'} \chi_\pi(k') > 0$. With this in mind, one can see that any boundary condition $G \to K'$ with $K' \neq G$ has some condensed irrep $\pi \neq 1$ as follows. For a regular representation, we get $\sum_{k' \in K'} \chi_{\mathrm{reg}}(k') = |G|$. When the regular rep is split into the irreps, the trivial rep contributes by $|K'|$ to the rhs. So when $|G| > |K'|$ there must be a nontrivial irrep $\pi$ with $\sum_{k' \in K'} \chi_\pi(k') > 0$, which must be condensed.

as a bicrossed product $G = H \bowtie K$, the $G/K$ symmetry in $D \geq 3$ dimensions with any twist $\omega_{D+1}$ always admits realization in symmetry-preserving gapped phases.

To see this, we again consider the symmetry TQFT where the bulk TQFT is given by the $G$ gauge theory with twist $\omega_{D+1}$. The symmetry-preserving gapped phase corresponds to the thin interval of the bulk gauge theory sandwich by a pair of gapped boundaries: one boundary breaks the gauge group $G \to K$, and the other breaks $G \to H$. Since $[\omega_{D+1}|_H]$ can still be nontrivial in general, the $G \to H$ boundary is realized by the $D$-dimensional non-invertible TQFT with $H$ symmetry carrying the anomaly $[\omega_{D+1}|_H]$. Such an anomalous TQFT can be generally obtained by a finite gauge theory through the group extension of $H$ by a finite gauge group [47, 48]. By shrinking the size of the interval, we get a symmetry-preserving gapped phase with $(H \bowtie K)/K$ symmetry with generic twist.

## 3.6   Dynamical scenario: symmetry-enforced gaplessness

In the previous section, we have seen that finite $(H \bowtie K)/K$ symmetry with generic twist can be realized in symmetry-preserving gapped phases. Here we study the inverse statement; if a finite group $G$ does not admit the expression in terms of bicrossed product $G = H \bowtie K$ with any subgroup $H \subset G$, the system preserving the coset symmetry $G/K$ must be gapless. We show this statement fully in general in spacetime dimensions $D \geq 3$.

### 3.6.1   Symmetry-enforced gaplessness: an example

As an example, let us explicitly see that the $A_5/\mathbb{Z}_2$ symmetry exhibits symmetry-enforced gaplessness. Suppose that there exists a gapped phase in $D$ spacetime dimensions with $A_5/\mathbb{Z}_2$ symmetry, where the $(D-2)$-form symmetry generated by the $\mathbb{Z}_2$ Wilson line is unbroken. We can gauge the $(D-2)$-form symmetry generated by the Wilson line, which brings the theory back to the gapped phase with $A_5$ symmetry. Then the dual $\mathbb{Z}_2 = \{1, s\}$ symmetry must be broken. So the $A_5$ symmetry must have a nontrivial SSB pattern to the subgroup $A_5 \to H$. Now there is no subgroup of $A_5$ with order $|A_5|/|\mathbb{Z}_2| = 30$, so there exists $g \in A_5$ with $g \neq s$ where $g, gs$ are broken. This implies that the 0-form coset non-invertible symmetry $\tilde{U}_g$ of the original theory is broken. This shows that the gapped phase with $A_5/\mathbb{Z}_2$ must spontaneously break either the $(D-2)$-form $\mathbb{Z}_2$ symmetry or the 0-form non-invertible symmetry, and thus leads to the symmetry-enforced gaplessness.

### 3.6.2   General statement about symmetry-enforced gaplessness

This argument can be extended to generic finite coset symmetry $G/K$, which may or may not be twisted. The argument for spacetime dimensions $D \geq 3$ is fully general.

In the case of $D = 2$, the statement is shown when any subgroup $K' \subset K$ has trivial 2nd group cohomology, $H^2(BK', U(1)) = 0$. For instance $K = \mathbb{Z}_N$ satisfies this property,

so $G/\mathbb{Z}_N$ symmetry in (1+1)D exhibits symmetry-enforced gaplessness when $G$ cannot be expressed as $H \rtimes \mathbb{Z}_N$.

Below, we consider a gapped system with $G/K$ symmetry preserving the $\mathrm{Rep}(K)$ Wilson lines, and assume that $G$ cannot be expressed as a bicrossed product $H \bowtie K$. We then show that some 0-form symmetry in $G/K$ must be broken, therefore $G/K$ symmetry cannot be fully preserved in gapped phases.

When $\mathrm{Rep}(K)$ $(D-2)$-form symmetry is preserved in $D \geq 3$, we claim that gauging $\mathrm{Rep}(K)$ symmetry always yields the dual $K$ symmetry fully broken spontaneously. To see this, suppose we obtain a phase where $K' \subset K$ symmetry is preserved. Gauging the $K$ symmetry back leads to finite group $K'$ gauge theory. In such theory above $D \geq 3$, the $K'$ Wilson lines are deconfined, and the $\mathrm{Rep}(K')$ subcategory symmetry is spontaneously broken. This implies that $K' = \{\mathrm{id}\}$. The same argument applies in $D = 2$ when $\mathrm{Rep}(K)$ has no nontrivial SPTs, i.e. $H^2(BK', U(1)) = 0$ for any subgroup $K' \subset K$.

Then, let us gauge the $(D-2)$-form symmetry of the $\mathrm{Rep}(K)$ Wilson lines, which leads to the phase with $G$ symmetry with $K \subset G$ fully broken. We can then express $G$ as $G = QK$, where $Q \subset G$ is a set of representatives from each element of the left coset of $K$ in $G$. Since $G$ is not a bicrossed product, one cannot find a choice of $Q$ such that $Q$ is a group. This is incompatible with the SSB pattern $G \to H$ with the symmetry group satisfying $|H| = |G|/|K|$ where one can have $Q = H$, so the unbroken subgroup $H$ has the order $|H| < |G|/|K|$. This implies that there exists $q \in G$ such that all of the group elements in $qK$ are broken. Now, the 0-form non-invertible symmetry in $G/K$ labeled by this $q \in G$ must be broken.[6] This shows that the coset symmetry with $G/K$ without an expression in terms of bicrossed product exhibits symmetry-enforced gaplessness in $D > 2$ and the same result holds for certain $K$ in $D = 2$.

### 3.6.3 Symmetry-enforced gaplessness of continuous coset symmetry

When the coset symmetry is continuous, certain anomalies imply the system must be gapless, similar to the symmetry-enforced gaplessness discussed in [7, 49, 50, 51]. We will focus on continuous coset symmetries with continuous $G$ and finite group $K$, where the discreteness of $K$ guarantees the coset symmetry defects described by the "sandwich construction" are topological defects and thus generate symmetries.

A large class of such anomalies comes from the anomalies of $G$ symmetry such that under a symmetry transformation the anomaly is nontrivial on spacetimes where TQFTs can only have positive partition functions, and thus any systems with such anomalous $G$ symmetry

---

[6]To see this, recall that the coset symmetry $\tilde{U}_q$ can be represented as a sandwiched operator $\tilde{U}_q = \mathcal{D}_{\mathrm{Rep}(K)} U_q \overline{\mathcal{D}}_{\mathrm{Rep}(K)}$ using a half-gauging defect $\mathcal{D}_{\mathrm{Rep}(K)}$ (see Figure 1). Taking trace of this operator within the low-energy Hilbert space gives $\mathrm{Tr}\left(\tilde{U}_q\right) = \mathrm{Tr}\left(U_q \overline{\mathcal{D}}_{\mathrm{Rep}(K)} \mathcal{D}_{\mathrm{Rep}(K)}\right) = \mathrm{Tr}\left(U_q(\sum_{k \in K} U_k)\right)$. If $qK$ are all broken, we obtain $\mathrm{Tr}\left(\tilde{U}_q\right) = 0$ on a sphere. This implies that $\tilde{U}_q$ is spontaneously broken.

must be gapless (if the continuous $G$ symmetry is spontaneously broken, the Goldstone modes are also gapless). If there were any gapped system with the corresponding coset symmetry, after gauging the finite $\text{Rep}(K)$ symmetry this would give another gapped system with anomalous $G$ symmetry, and we would have a contradiction. Therefore any system with such anomalous coset symmetry must be gapless.

For example, take $D = 4$ and $G = SU(2)$ with $\omega_5$ given by Witten anomaly [52],[7] and $K$ given by any finite subgroup of $SU(2)$. Such $K$ is non-anomalous since finite group symmetry does not have nontrivial instanton number on $S^4$, while Witten anomaly means that there is an odd number of fermion zero modes in the presence of background with minimal $SU(2)$ instanton number on $S^4$ [52]. Suppose there were any gapped system with such anomalous coset symmetry, then by gauging $\text{Rep}(K)$ we would recover a gapped system with anomalous $G$ symmetry. However, such system would be forbidden by Witten anomaly, since a fermion parity $(-1)^F$ transformation changes the sign of the partition function on $S^4$ with $SU(2)$ instanton background and thus the partition function vanishes, which cannot happen for TQFT partition functions on $S^4$. Therefore we conclude that any system with such anomalous coset symmetry must be gapless.

## 3.7 Dynamical scenario: gapped phases with broken symmetries

### 3.7.1 Gapped phases with unbroken 0-form symmetry but broken $(D-2)$-form symmetry

Let us also describe another dynamical scenario where the system is gapped and symmetric under 0-form symmetry, but the $(D-2)$-form symmetry is spontaneously broken. In $D \geq 3$, any finite coset symmetry can be realized in gapped phases under such symmetry breaking.

To see this, we note that anomalous finite group $G$ symmetry in $D \geq 3$ spacetime dimensions with anomaly described by group cohomology or beyond group cohomology bosonic anomalies can always be realized by symmetric gapped phases [47, 54]. Since in the coset symmetry $(G, K, \omega_{D+1}, \alpha_D)$, the subgroup $K$ is anomaly-free $\omega_{D+1}|_K = d\alpha_D$, we can gauge the subgroup $K$ symmetry in such symmetric gapped phase with anomalous $G$ symmetry. As $K$ is a finite group, this produces another symmetric gapped phase, and thus we construct a symmetric gapped phase with the coset symmetry.

---

[7]Witten anomaly is a fermionic anomaly which is not described by group cohomology, but rather expressed as the topological term $\eta \cup p_1(SU(2))$ for spin structure $\eta$ and $p_1(SU(2))$ is the first Pontryagin class of $SU(2)$ bundles describing the instanton number of $SU(2)$. Hence, the coset symmetry presented here involves the twist by the spin invertible phase, which is a fermionic generalization of the twisted coset symmetry $(G, K, \omega_{D+1}, \alpha_D)$ discussed in this paper. The topological term for the inflow of Witten anomaly corresponds to the Gu-Wen supercohomology SPT phase in (4+1)D, characterized by the fourth cohomology class $H^4(BSU(2), \mathbb{Z}_2) = \mathbb{Z}_2$ generated by the mod 2 reduction of the Pontryagin class $p_1(SU(2))$ [53].

### 3.7.2   Gapped phases with broken 0-form symmetries

Let us consider the dynamical scenario where the 0-form discrete coset symmetry in $(G, K, \omega_{D+1}, \alpha_D)$ is spontaneously broken. What is the constraint on the vacua on sphere in $D \geq 2$? Note that since the Wilson lines can be contracted on spheres, $\mathrm{Rep}(K)$ acts trivially on spheres and is unbroken, and similarly the symmetry generated by the condensation defects of $\mathrm{Rep}(K)$ is also unbroken. Thus the number of vacua is given by spontaneously broken $G$ symmetry vacua identified by the $K$ gauge symmetry:

$$\#\text{vacua} \in \frac{|G|}{|K|}\mathbb{Z} \ . \tag{3.4}$$

**Description using bulk TQFT**   The spontaneously 0-form symmetry breaking phase can also be described using the bulk TQFT, where the vacua are described by topological local operators given by Wilson lines stretching between the boundary with coset symmetry and the other reference gapped boundary (here, let us consider $D \geq 3$ so the magnetic flux excitations in the bulk are not point-like). The minimal nontrivial number of vacua realizing anomalous coset symmetry is given by the minimal number of nontrivial irreducible representations of $G$ whose decompositions under the subgroup $K$ and another subgroup $K'$ contain the identity.

As an example, consider spontaneously broken coset symmetry $(G = S_3 = \mathbb{Z}_3 \rtimes \mathbb{Z}_2, K = \mathbb{Z}_2, \omega_{D+1} = 0, \alpha_D = 0)$. For the coset symmetry to be broken, we choose the reference boundary to be the $e$-condensed boundary, where the Wilson lines are the irreducible representations 1, 1-dimensional $\mathbb{Z}_2$ Wilson line $W$, and the 2-dimensional representation $\sigma$ that combines the charge-$(\pm 1)$ representations of $\mathbb{Z}_3$. The Wilson lines that can extend between the two boundaries are those that decompose under $\mathbb{Z}_2$ contains the identity, and they are $1, \sigma$. Thus there are $1 + 2 = 3$ point operators (counting with dimension), which give rise to 3 vacua, in agreement with (3.4).

## 4   Lattice Model with Anomalous Coset Symmetry

Here we study lattice models of finite gauge theory with anomalous or non-anomalous coset symmetry. We will see that the anomalies can forbid the Higgs phases of the gauge theory preserving the coset symmetry.

### 4.1   Non-anomalous case: $K$ gauge theory with $(H \bowtie K)/K$ symmetry

Let us consider a $K$ gauge theory with untwisted $(H \bowtie K)/K$ symmetry in generic $D$ spacetime dimensions. The lattice model can be described in generic spatial dimensions, but

let us work on 2d space $(D = 3)$ for simplicity. We start with the trivial gapped phase with $G = H \bowtie K$ symmetry,

$$H_{H \bowtie K} = -\sum_v \left( \sum_{g \in H \bowtie K} \overrightarrow{X}_v^g \right).$$ 

(4.1)

Since $G = H \bowtie K$ is a bicrossed product, each state $|g\rangle$ with $g \in G$ can be uniquely expressed by a pair $|g\rangle = |h, k\rangle$, $h \in H, k \in K, g = hk$. The theory after gauging $K$ is given by

$$H_{(H \bowtie K)/K} = -\sum_v \left( \sum_{g=(h,k)} \overrightarrow{X}_v^g \right) - \sum_p B_p \,,$$ 

(4.2)

with the exact $K$ Gauss law constraint

$$\overrightarrow{X}_v^{(1,k)} A_v^k = 1.$$ 

(4.3)

Here we defined the group-based Pauli $X$ like operators as

$$\overrightarrow{X}^g |h\rangle = |gh\rangle, \quad \overleftarrow{X}^{g^{-1}} |h\rangle = \left| hg^{-1} \right\rangle .$$ 

(4.4)

$A_v^k, B_p$ are the Hamiltonian terms of the quantum double model with the gauge group $K$

$$A_v^k = \overrightarrow{X}_{N(v)}^k \overrightarrow{X}_{E(v)}^k \overleftarrow{X}_{W(v)}^{k^{-1}} \overleftarrow{X}_{S(v)}^{k^{-1}}, \quad B_p = \delta_{k_{01} k_{13} k_{02}^{-1} k_{23}^{-1}, 0} \,.$$ 

(4.5)

These terms are shown in Figure 5. This Hamiltonian has the coset $(H \bowtie K)/K$ symmetry generated by the operator

$$\tilde{U}_g = \Pi \mathcal{D} \left( \prod_v \overrightarrow{X}_v^g \right) \mathcal{D} \Pi$$ 

(4.6)

where $\Pi$ is the projection onto the $K$ Gauss law, and $\mathcal{D}$ is the projection onto the trivial $K$ gauge field

$$\mathcal{D} = \prod_e \delta_{1, k_e}$$ 

(4.7)

Its fusion rule is given by

$$\tilde{U}_g \times \tilde{U}_{g'} = \sum_{k \in K} \tilde{U}_{gkg'k^{-1}}$$ 

(4.8)

Note that when $g = hk$, $\tilde{U}_g = \tilde{U}_h$. With this in mind, one can consider a perturbation which preserves the coset $(H \bowtie K)/K$ symmetry. The perturbation is given by the Hamiltonian

$$V = -\sum_{e=\langle vv' \rangle} \delta_{k_v^{-1} k_e k_{v'}}$$ 

(4.9)

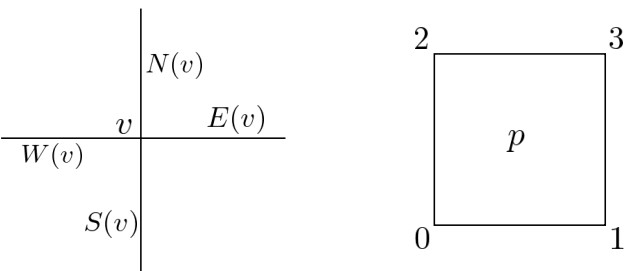

Figure 5: The edges nearby a vertex $v$ and a plaquette $p$.

where $g_v = h_v k_v, g_{v'} = h_{v'} k_{v'}$. This perturbation commutes with the Gauss law. This also commutes with the coset symmetry since $V$ commutes with the operator $\overrightarrow{X}_h$ with $h \in H$.

The perturbed Hamiltonian

$$H^0_{(H \bowtie K)/K} = -\sum_v \left( \sum_{g=(h,k)} \overrightarrow{X}^g_v \right) - \sum_{e=\langle vv' \rangle} \delta_{k_v^{-1} k_e k_{v'}} \tag{4.10}$$

with the $K$ Gauss law is a trivial gapped phase preserving the coset symmetry. This phase is regarded as the Higgs phase of the $K$ gauge theory with the electric charges condensed, where the ground state is given by an SPT phase with $(H \bowtie K)/K$ symmetry. The ground state is given by

$$|\text{GS}\rangle = \sum_{\{g_v\}} \bigotimes_v |g_v\rangle \bigotimes_{e=\langle vv' \rangle} \left| k_v k_{v'}^{-1} \right\rangle \tag{4.11}$$

This is consistent with the fact that the $(H \bowtie K)/K$ coset symmetry is non-anomalous.

### 4.1.1 SPT phases with $(H \bowtie K)/K$ symmetry

The Hamiltonian (4.10) gives an example of SPT phases with the coset $(H \bowtie K)/K$ symmetry. The other SPT phases with the same coset symmetry can be obtained by using the disentanglers of the $H$ SPT phases. To see this, let us consider the trivial $H$ SPT phase on a lattice with the local Hilbert space $\{|h\rangle, h \in H\}$ on each vertex,

$$H^0_H = \sum_v \left( \sum_{h \in H} \overrightarrow{X}_h \right), \tag{4.12}$$

and then the generic $H$ SPT labeled by $\omega \in H^D(BH, U(1))$ can be obtained by conjugating $H^0_H$ by a finite depth circuit $U_\omega$, $H^\omega_H = U_\omega H^0_H U_\omega^\dagger$. A subclass of SPT phases with $(H \bowtie K)/K$ is also labeled by the group cohomology $\omega \in H^D(BH, U(1))$. The distinct SPT phases are then given by $H^\omega_{(H \bowtie K)/K} = U_\omega H^0_{(H \bowtie K)/K} U_\omega^\dagger$, where $U_\omega$ acts on vertices through $\{h_v\}$ with $g_v = h_v k_v$.

## 4.2   Anomalous case: $\mathbb{Z}_2$ gauge theory with $A_5/\mathbb{Z}_2$ symmetry

Here we study the $\mathbb{Z}_2$ gauge theory with the coset $A_5/\mathbb{Z}_2$ symmetry. We start with the trivial gapped phase with $A_5$ symmetry

$$H_{A_5} = -\sum_v \left( \sum_{g \in A_5} \overrightarrow{X}_v^g \right) \tag{4.13}$$

Let us write a generator of $\mathbb{Z}_2$ as $s = (1,2)(3,4)$. One can take a subset $Q \subset A_5$ with $|Q| = 30$ such that $A_5 = Q \sqcup Qs$, $Q = Q^{-1}$, and $\mathrm{id} \in Q$. The set $Q$ with such properties satisfies $sQ = Qs, sQs = Q$. Note that $Q$ is not a group, since $A_5$ is simple and does not have a normal subgroup, or simply from the fact that $A_5$ does not have a subgroup with order 30. To gauge the $\mathbb{Z}_2$ symmetry, we introduce a single qubit on each edge for the $\mathbb{Z}_2$ gauge field. Then the theory after gauging the $\mathbb{Z}_2$ symmetry is expressed as

$$H_{A_5/\mathbb{Z}_2} = -\sum_v \left( \sum_{g \in A_5} \overrightarrow{X}_v^g \right) - \sum_p B_p \tag{4.14}$$

where $B_p$ is the plaquette operator

$$B_p = \prod_{e \subset \partial p} Z_e \tag{4.15}$$

The $\mathbb{Z}_2$ Gauss law constraint is given by

$$\overrightarrow{X}_v^s A_v = 1. \tag{4.16}$$

The gauged Hamiltonian $H_{A_5/\mathbb{Z}_2}$ commutes with the $\mathbb{Z}_2$ Gauss law. This Hamiltonian has the coset $A_5/\mathbb{Z}_2$ symmetry

$$\tilde{U}_g = \Pi \mathcal{D} \left( \prod_v \overrightarrow{X}_v^g \right) \mathcal{D} \Pi \tag{4.17}$$

where $\Pi$ is the projection onto the $\mathbb{Z}_2$ Gauss law, and $\mathcal{D}$ is the projection onto the trivial $\mathbb{Z}_2$ gauge field

$$\mathcal{D} = \prod_e \left( \frac{1 + Z_e}{2} \right) \tag{4.18}$$

Its fusion rule is given by

$$\tilde{U}_g \times \tilde{U}_{g'} = \tilde{U}_{gg'} + \tilde{U}_{gsg's} \tag{4.19}$$

Note that when $g = qs$, we have $\tilde{U}_g = \tilde{U}_q$, so the symmetry operator can be labeled by the element $q \in Q$.

Due to the 't Hooft anomaly, anyon condensation of $\mathbb{Z}_2$ gauge theory to the trivial gapped phase is expected to violate the coset $A_5/\mathbb{Z}_2$ symmetry. Since $\mathcal{D}$ projects out the states with the $\mathbb{Z}_2$ magnetic fluxes, one cannot condense magnetic fluxes while preserving the coset symmetry. To see how the symmetry forbids the condensation of electric charges, let us consider the following gauge invariant term condensing the charges

$$V = - \sum_{e=\langle vv'\rangle} Z_v Z_e Z_{v'} \tag{4.20}$$

where we define $Z_v$ as $Z_v |q\rangle = |q\rangle$, $Z_v |qs\rangle = -|qs\rangle$ for $q \in Q$. This operator commutes with the $\mathbb{Z}_2$ Gauss law, but does not commute with the coset symmetry $\tilde{U}_g$. To see this, let us recall that $Q$ is not a group and $Q$ is not closed under the left $q \in Q$ action. This implies that $\overrightarrow{X}_v^q Z_v = Z_v \overrightarrow{X}_v^q$ is not satisfied by all $q \in Q$, so the commutation fails for some $\tilde{U}_q$ with $q \in Q$. This is in contrast to the $K$ gauge theory with non-anomalous $(H \rtimes K)/K$ symmetry, where we could condense electric charges to bring the theory into a trivial gapped phase, while preserving the coset symmetry.

# 5 Discussion and Outlook

## 5.1 Comment on obstruction to gauging

In this paper, we mainly focused on the 't Hooft anomalies of coset symmetries as obstruction to SPT phases. Meanwhile, there is another definition of 't Hooft anomaly as whether one can gauge the symmetry. These two definitions of anomalies are equivalent for invertible internal symmetries, but bifurcate for non-invertible symmetries. For non-invertible symmetries, the obstruction to gauging and obstruction to SPT phases are generally different. For example, the non-invertible symmetry given by Fibonacci fusion rules in (1+1)D cannot be realized in a trivially gapped phase, but the Fibonacci fusion category has an algebra object containing the non-invertible object and can be gauged [55].

For general non-invertible symmetry, gauging the symmetry means inserting "mesh" of symmetry defects, whose consistency relies on existence of an algebra object in the symmetry category. For a given symmetry category, it is in general difficult to enumerate all of the possible algebra objects. In the case of finite coset symmetry without twist $(G, K, 0, 0)$, one can always gauge the $G/K$ symmetry via sequential gauging; we first gauge the symmetry generated by $\mathrm{Rep}(K)$ Wilson lines, and then gauge the $G$ symmetry. This sequential gauging corresponds to an algebra object of the symmetry category with the form of

$$\mathcal{A}_{G/K} = \mathcal{D}_{\mathrm{Rep}(K)} \times \left( \bigoplus_{g \in G} U_g \right) \times \overline{\mathcal{D}}_{\mathrm{Rep}(K)} \ , \tag{5.1}$$

up to overall normalization factor. Here $\mathcal{D}_{\mathrm{Rep}(K)}$ is a half gauging of $\mathrm{Rep}(K)$ Wilson lines, and $U_g$ is the $g \in G$ symmetry defect of the theory after gauging Wilson lines. Inserting a fine mesh of $\mathcal{A}_{G/K}$ yields the theory after the sequential gauging.

We note that the algebra object $\mathcal{A}_{G/K}$ becomes degenerate in general, in the sense of $\langle \mathcal{A}_{G/K}, 1 \rangle > 1$ with $\langle \mathcal{A}, x \rangle := \dim(\mathrm{Hom}(\mathcal{A}, x))$. The algebra object with $\langle \mathcal{A}_{G/K}, 1 \rangle = 1$ is called a haploid algebra, so $\mathcal{A}_{G/K}$ with generic finite untwisted coset symmetry may or may not be haploid. For instance, let us consider the anomalous fusion category symmetry $A_5/\mathbb{Z}_2$ in (1+1)D. In this case $\mathcal{A}_{A_5/\mathbb{Z}_2}$ has $\langle \mathcal{A}_{A_5/\mathbb{Z}_2}, 1 \rangle = 2$. Also, the algebra object $\mathcal{A}_{A_5/\mathbb{Z}_2}$ does not factorize as $\mathcal{A}_{A_5/\mathbb{Z}_2} = 2\mathcal{A}'$ with a haploid algebra object $\mathcal{A}'$. [8] Therefore the algebra object for the sequential gauging of $A_5/\mathbb{Z}_2$ is not haploid. Given that the existence of haploid algebra object has implications for gauging the symmetry [55], it would be interesting to find such physical applications of non-haploid algebra objects.

## 5.2    Future directions

There are a number of future directions:

**Dynamical consequences and concrete realizations**    It would be interesting to find relations between the coset symmetry data $(G, K, \omega_{D+1}, \alpha_D)$ and experimentally measurable quantities. For example, when $\omega_{D+1}$ is an exact cocycle, it is related to topological responses.

In addition, it would be interesting to find more dynamical applications for coset symmetries in simple models. For example, we can explore phase transitions in lattice models with coset symmetries [12].

A large class of systems with coset symmetry comes from Higgs phase of gauge theories, where a gauge group $G$ is broken to a subgroup $K$ that remains gauged. Such system necessarily have coset symmetry with trivial twist $\omega_{D+1} = 0$, since the original system must be free of gauge anomaly. In a companion paper we will discuss these applications in more details.

**Generalization to fermionic systems**    While the coset symmetry studied here is described by $G$ gauge theory in the bulk with boson Wilson line, it would be interesting to study "fermionic coset symmetry" where the bulk is a gauge theory with emergent fermions. This includes the cases that the symmetry line operators included in the coset symmetry are not bosons (unlike those in $\mathrm{Rep}(K)$ in the coset symmetry discussed in the paper). For example, this occurs when we allow the presence of a fermionic lines that are Wilson lines of fermion

---

[8]This can be seen as follows. Since $\mathcal{A}_{A_5/\mathbb{Z}_2}$ has total quantum dimension $\dim(\mathcal{A}_{A_5/\mathbb{Z}_2}) = 2|A_5| = 120$, such $\mathcal{A}'$ would satisfy $\dim\mathcal{A}' = |A_5| = 60$. This total dimension is maximal in the sense that for any simple $x$ we must have $\langle \mathcal{A}', x \rangle = d_x$ with $d_x$ the quantum dimension of $x$. However, since $A_5/\mathbb{Z}_2$ does not admit the realization in SPT phases, $A_5/\mathbb{Z}_2$ does not have such a haploid algebra object with the maximal total quantum dimension [55].

parity. We can also explore potential generalizations of coset symmetries that can only exist due to presence of physical fermions.

**Relation to higher fusion categories**  In general spacetime dimension $D$, the coset symmetry describes a fusion $(D-1)$-category. Higher fusion categories are mysterious and it would be interesting to use the coset symmetry as specified by the data $(G, K, \omega_{D+1}, \alpha_D)$ to explore properties of higher fusion categories such as their consistency conditions for fusion and braiding. For example, consistency condition on fusion can originate from the cocycle conditions for $\omega_{D+1}$.

When $D = 3$, the coset symmetry describes a fusion 2-category [32], and it would be interesting to compare the data $(G, K, \omega_4, \alpha_3)$ with the fusion 2-category classification [56]. In particular, while we show that $\alpha_D$ can change the algebraic structure of the symmetry including its fusion rule, it would be interesting to further explore its role.

**Explore variety of theories related by gauging coset symmetries**  While the focus of the paper is obstruction to SPT phases, it can be interesting to discuss the obstruction to gauging as discussed partially in section 5.1, and also the relation between theories related by gauging coset symmetries. For example, quantities like critical exponents in theories related by gauging a discrete symmetries are the same.

# Acknowledgment

We thank Yichul Choi, Kansei Inamura, Sakura Schafer-Nameki and Matthew Yu for helpful discussions. P.-S.H. is supported by Department of Mathematics King's College London. R.K. is supported by the U.S. Department of Energy through grant number DE-SC0009988 and the Sivian Fund. C.Z. is supported by the Harvard Society of Fellows and the Simons Collaboration on Ultra Quantum Matter. The authors of this paper were ordered alphabetically.

# A   Anomaly-Free Condition for $(H \bowtie K)/K$ Coset Symmetries

We will discuss the untwisted (i.e. $\omega_{D+1} = 0, \alpha_D = 0$) coset symmetry with $G = H \bowtie K$ for some subgroup $H$, and the $K$ subgroup is gauged. We show that there are no irrep $\rho \in \text{Rep}(G)$ such that its decompositions under the subgroups $H, K$ simultaneously contains the trivial representations of $H, K$. We will prove it by contradiction.

Suppose there exists such irrep $\rho$ acting on the vector space $V$, which decomposes into $\rho = 1 \oplus \dots$ under $H, K$. Since the decomposition under $K$ contains a trivial representation, one can take a state $|\psi\rangle$ satisfying $\rho(k) |\psi\rangle = |\psi\rangle$ for any $k \in K$.

Then, let us take a vector space $\tilde{V} \subset V$ spanned by $\rho(h) \ket{\psi}$ with any $h \in H$. One can immediately check that irrep $\rho$ acts within $\tilde{V}$. Pick some state $\rho(h') \ket{\psi} \in \tilde{V}$ with $h' \in H$. For $g = hk \in G, h \in H, k \in K$,

$$\rho(g) \cdot \rho(h') \ket{\psi} = \rho(h)\rho(k)\rho(h') \ket{\psi} = \rho(hh'') \ket{\psi} \in \tilde{V}, \tag{A.1}$$

with $kh' = h''k''$ for some $h'' \in H, k'' \in K$. Since $\rho$ is an irrep, we get $V = \tilde{V}$. This implies any state of $V$ is expressed as a linear combination of states in the form of $\rho(h) \ket{\psi}$.

Now since the decomposition of $\rho$ contains a trivial representation under $H$, there exists a non-vanishing state $\ket{\lambda} = \sum_{h \in H} \lambda_h \rho(h) \ket{\psi}$ satisfying $\rho(h) \ket{\lambda} = \ket{\lambda}$ for any $h \in H$. This implies

$$\ket{\lambda} = \frac{1}{|H|} \sum_{h \in H} \rho(h) \ket{\lambda} = \left( \frac{1}{|H|} \sum_{h' \in H} \lambda_{h'} \right) \left( \sum_{h \in H} \rho(h) \right) \ket{\psi} \propto \left( \sum_{h \in H} \rho(h) \right) \ket{\psi} \tag{A.2}$$

So, one can take $\lambda_h = 1$ for any $h \in H$ for the definition of $\ket{\lambda}$. One can check that $\ket{\lambda}$ is invariant under the action of $k \in K$. We have

$$\rho(k) \ket{\lambda} = \sum_{h' \in H} \rho(kh') \ket{\psi} \tag{A.3}$$

One can immediately see that $kh_1$ and $kh_2$ with $h_1 \neq h_2$ leads to the expression $kh_i = h_i' k_i'$ with distinct elements $h_1' \neq h_2'$ of $H$. [9] This implies

$$\rho(k) \ket{\lambda} = \sum_{h' \in H} \rho(kh') \ket{\psi} = \sum_{h \in H} \rho(h) \ket{\psi} = \ket{\lambda} \tag{A.4}$$

Then, this state $\ket{\lambda}$ realizes the 1d irrep of $G$. For $g = hk \in G$,

$$\rho(g) \ket{\lambda} = \rho(h)\rho(k) \ket{\lambda} = \ket{\lambda} . \tag{A.5}$$

This is in contradiction with $\rho$ is an irrep of $G$, which completes the proof.

# B    Anomalous $A_6/A_5$ Coset Symmetry

Let us provide another example of anomalous finite coset symmetry without a twist. Consider an alternating group $G = A_6$ and its subgroup $K = A_5$. Since $A_6$ is simple, $A_5$ is not a normal subgroup and $A_6/A_5$ is non-invertible. $A_5, A_6$ do not have a common normal subgroup so this is a minimal coset symmetry.

Also, $A_6$ cannot be expressed as a bicrossed product of two subgroups $H, K$ with $K = A_5$. This can be verified by checking that every subgroup $H$ of order $6 = |A_6|/|A_5|$ has a nontrivial

---

[9]Let us check the contraposition. Suppose that $kh_1 = hk_1, kh_2 = hk_2$ with $h \in H$. This implies $k_1 h_1 = k_2 h_2$, so $h_1 h_2^{-1} = k_1^{-1} k_2$. Since $H \cap K = \{\text{id}\}$ we get $h_1 = h_2$.

| | $C_1$ | $C_2$ | $C_3$ | $C_4$ | $C_5$ | $C_6$ | $C_7$ |
|---|---|---|---|---|---|---|---|
| Element | () | $(1,2)(3,4)$ | $(1,2,3)$ | $(1,2,3)(4,5,6)$ | $(1,2,3,4)(5,6)$ | $(1,2,3,4,5)$ | $(1,2,3,4,6)$ |
| $|C_i|$ | 1 | 45 | 40 | 40 | 90 | 72 | 72 |

Table 1: Conjugacy classes of $A_6$ that corresponds to the character table.

overlap with $A_5$. So $A_6/A_5$ is a candidate of the coset non-invertible symmetry that does not admit realization in SPT phases.

Let us consider the symmetry TQFT for the coset $G/K$ symmetry, which is the $G = A_6$ gauge theory. One boundary breaks the gauge group $G \to K$, and the other breaks $G \to K'$. If $|K'| \geq 6$, $K \cap K' \neq \{\text{id}\}$. This implies the presence of a nontrivial magnetic operator stretching between the ends of the interval, which leads to the nontrivial degeneracy of states. To find SPT we need to look for the cases with $|K'| \leq 5$. Writing $K = A_5 = \langle(1,2,3,4,5),(1,2,3)\rangle$, the choices of the nontrivial subgroup $K' \subset A_6$ with $K \cap K' = \{\text{id}\}$ are listed as follows (up to the permutation isomorphism):

$$\begin{aligned}
\mathbb{Z}_2 &= \langle(1,2)(3,6)\rangle, \quad A_3 = \mathbb{Z}_3 = \langle(1,2,6)\rangle, \\
A_3^{\text{diag}} &= \mathbb{Z}_3 = \langle(),(1,2,3)(4,5,6),(1,3,2)(4,6,5)\rangle, \\
\mathbb{Z}_2^2 &= \langle(),(1,2)(3,6),(1,3)(2,6),(1,6)(2,3)\rangle, \\
\mathbb{Z}_4 &= \langle(),(1,2,5,6)(3,4),(1,5)(2,6),(1,6,5,2)(3,4)\rangle, \\
\mathbb{Z}_5 &= \langle(1,2,3,4,6)\rangle .
\end{aligned}$$
(B.1)

Let us study if the interval realizes an SPT with these groups $K'$ listed above. It is useful to explicitly write down the character table of $A_6$ (generated by GAP):

$$\begin{pmatrix}
1 & 1 & 1 & 1 & 1 & 1 & 1 \\
5 & 1 & 2 & -1 & -1 & 0 & 0 \\
5 & 1 & -1 & 2 & -1 & 0 & 0 \\
8 & 0 & -1 & -1 & 0 & -\omega_5 - \omega_5^4 & -\omega_5^2 - \omega_5^3 \\
8 & 0 & -1 & -1 & 0 & -\omega_5^2 - \omega_5^3 & -\omega_5 - \omega_5^4 \\
9 & 1 & 0 & 0 & 1 & -1 & -1 \\
10 & -2 & 1 & 1 & 0 & 0 & 0
\end{pmatrix} ,$$
(B.2)

where $\omega_5$ is the 5th root of unity. Each row corresponds to an irrep $R_1, \ldots, R_7$, and each column corresponds to a conjugacy class $C_1, \ldots, C_7$. The representative and the size of each conjugacy class is given in Table 1.

We will see that for $K'$ listed above, the 5d irrep $R_2$ contains a trivial rep under both $K = A_5, K'$. In general, $G$ irrep $R$ contains direct sum of $d$ trivial reps under $H \subset G$ if

$$\sum_{h \in H} \chi_R(h) = d|H| .$$
(B.3)

The sum of characters can be directly evaluated for $H = K, K', R = R_2$ using the character table as follows:

$$\sum_{k \in A_5} \chi_{R_2}(k) = \chi_{R_2}(C_1) + 15\chi_{R_2}(C_2) + 20\chi_{R_2}(C_3) + 12\chi_{R_2}(C_6) + 12\chi_{R_2}(C_7) = 60 \ ,$$

$$\sum_{k \in \mathbb{Z}_2} \chi_{R_2}(k) = \chi_{R_2}(C_1) + \chi_{R_2}(C_2) = 6 \ ,$$

$$\sum_{k \in A_3} \chi_{R_2}(k) = \chi_{R_2}(C_1) + 2\chi_{R_2}(C_3) = 9 \ ,$$

$$\sum_{k \in A_3^{\mathrm{diag}}} \chi_{R_2}(k) = \chi_{R_2}(C_1) + 2\chi_{R_2}(C_4) = 3 \ , \tag{B.4}$$

$$\sum_{k \in \mathbb{Z}_2^2} \chi_{R_2}(k) = \chi_{R_2}(C_1) + 3\chi_{R_2}(C_2) = 8 \ ,$$

$$\sum_{k \in \mathbb{Z}_4} \chi_{R_2}(k) = \chi_{R_2}(C_1) + \chi_{R_2}(C_2) + 2\chi_{R_2}(C_5) = 4 \ ,$$

$$\sum_{k \in \mathbb{Z}_5} \chi_{R_2}(k) = \chi_{R_2}(C_1) + 4\chi_{R_2}(C_7) = 5 \ .$$

Hence the number of trivial reps under $K, K'$ is always positive, $d > 0$. This implies that $R_2$ can terminate at both ends of the interval for any choices of $K'$ listed above. So the $A_6/A_5$ coset symmetry admits no realization in SPT phases.

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
