# Peer review of "Anomalies of Coset Non-Invertible Symmetries"

_SciPost Physics_

## Round 1 · Referee Report · Anonymous (Referee 1) · 2025-5-2

Report

This paper studies twisted coset symmetries, i.e. symmetries obtained by gauging a (non-normal) subgroup $K$ of a group $G$, possibly in presence of a non-trivial twist $\omega$, often resulting in a non-invertible symmetry. In particular, the authors study anomalies of coset symmetries, where the anomaly is intended as the obstruction to realizing such symmetry in an SPT phase. The authors explore also anomalous scenarios that lead to non-trivial phases, such as gapless phases and phases that spontaneously break the symmetry. Moreover, several concrete realizations are discussed both in the continuum and in lattice models.

Let me list some comments and questions:

  1. Second bullet point on pg. 4: reading later on it seems $\alpha_D$ can modify also the fusion rules of the symmetry;
  2. When the sandwich construction for twisted coset symmetry defects is explained on pg. 7, it is stated that this works for a large class of symmetry defects. It would be interesting to mention some instances where this construction is not applicable;
  3. Middle of page 12, before subsection 2.2.2.: why is it 2Rep$(\mathbb{Z}_2)$ when the boundary is 2D?;
  4. Bottom line of pg. 12: should it be the magnetic defect for $a_2$ instead of the magnetic defect for $a_1$?;
  5. Eq 2.29 has $\omega_4 = d \eta_3/2$, while in the introduction at the bottom of pg. 5 there is $\omega_4 = d \eta_3'/2$;
  6. Point (1) on pg. 19: why the subgroups need to satisfy $[\omega_{D+1}|_{K,K'}]=1$? The above condition for a gapped boundary is $[\omega_{D+1}|_{K,K'}]=0$. Moreover the bullet point (1) seems redundant since a gapped boundary is already labelled by a subgroup $K$ such that $[\omega_{D+1}|_{K}]=0$;
  7. In the previous paper by the authors [10], it is stated that the untwisted $S_3 / \mathbb{Z}_2$ coset symmetry cannot be realized in a trivially gapped theory (pg. 31). However $S_3 = \mathbb{Z}_3 \rtimes \mathbb{Z}_2$ and $\mathbb{Z}_3 \cap \mathbb{Z}_2 = { \text{id}}$, so this seems in contradiction with the conditions in section 3.2;
  8. Bottom of pg. 20: instead of "$G_1$ needs to break to subgroup of $K_1$", "$G_1$ needs to break to a subgroup $K_1$";
  9. In section 3.5, I find the term symmetric gapped phase slightly confusing, as this sometimes includes gapped phases that spontaneously break the symmetry, where this is realized as some permutation of the vacua. Also why $[\omega_{D+1}|_{H}]$ can still be non-trivial in general if a gapped boundary is labelled by a $H$ such that $[\omega_{D+1}|_{H}]=0$?;
  10. In middle of pg. 31, the sentence "consistency condition on fusion can from the cocycle conditions" is probably missing a "come" before "from";
  11. There are some grammar typos around the text that should be fixed, e.g. in the introduction "$K \subset G$ is a not a normal subgroup", etc.

Overall, I believe this paper is well-written and contains original and important results. Moreover the discussion is for arbitrary $D$ space-time dimensions and therefore very general and widely applicable, even though this may come sometimes at the cost of less concrete details. Therefore I recommend the paper for publication provided the above comments are taken into account.

Recommendation

Ask for minor revision

  • validity: -
  • significance: -
  • originality: -
  • clarity: -
  • formatting: -
  • grammar: -

Author:  Ryohei Kobayashi  on 2025-11-05  [id 5996]

(in reply to Report 1 on 2025-05-02)

  1. We thank the referee for pointing out this error. We edited this sentence to say that $\alpha_D$ can modify the fusion rules.

  2. If $K$ is continuous, defects won't be topological. We wrote a ``large class of defects'' since sometimes the sandwich defect is not simple, and one cannot construct all symmetry operators from the sandwich construction. For instance, this happens when $K$ is a normal subgroup of $G$; then $G/K$ is invertible but the sandwich construction gives a non-invertible operator which is not simple.

  3. We thank the referee for catching this. This was a typo and is now fixed to Rep($\mathbb{Z}_2$).

  4. In the example in section 2.2.2, the fusion of magnetic defects for $a_2$ is extended by the gauged SPT defect (reducing the $\omega_5$ on circle with $a_2$ holonomy).

  5. We have replaced the previous example with a better example of $O(2)/\mathbb{Z}_2$ symmetry, with $\omega$ is the theta term corresponds to the $O(2)_{1/2,1/2}$ fractional Chern-Simons term and $\alpha=\frac{1}{2}\eta$ being half of the minimal $\mathbb{Z}_2$ Chern-Simons term.

  6. We thank the referee for catching the typo. We use the convention that $[\omega_{D+1}]$ has coefficient in $\mathbb{R}/(2\pi\mathbb{Z})$, i.e. real number mod $2\pi$ multiple of integer, where the identity element under addition is 0 instead of 1. So indeed the equation should be $[\omega_{D+1}|_{K,K'}]=0$.

  7. That was the incorrect statement in [10]; S3/Z2 symmetry admits a realization in SPT phases. We updated the arXiv version of [10] and now there is no inconsistency.

  8. Thank you for the catch, we fixed it.

  9. Here we mean a symmetry-preserving gapped phases, hopefully the discussions in the main text makes this terminology clear. While in Sec.3.2 we considered the case with $\omega$ trivial within H, here we do not assume this. Therefore the H symmetry can be anomalous, and an anomaly of finite group symmetry is always saturated by symmetry-preserving gapped phases.

10, 11. Thanks for the catch, we fixed these typos.

---

## Round 1 · Referee Report · Anonymous (Referee 2) · 2025-6-9

Strengths

Concrete examples and comprehensive technical findings

Weaknesses

  1. The manuscript’s literature review is weak

  2. The technical sections are not well organized and are difficult to follow

Report

The manuscript presents interesting results and examples concerning the structure of coset non-invertible symmetries in general spacetime dimensions. These symmetries arise from the “dual symmetry,” obtained by gauging a finite non-normal subgroup of a global symmetry. The authors also examine anomalies associated with such symmetries. However, before recommending the manuscript for publication, the authors should address the following major concerns:

Requested changes

  1. The manuscript suggests that coset symmetry arises from starting with a global symmetry group $G$ possessing a ’t Hooft anomaly classified by $[\omega_{D+1}] \in H^{D+1}(G, U(1))$, and subsequently gauging an anomaly-free subgroup $K \subset G$ with a discrete torsion inside $H^D(K, U(1))$. The authors need to clearly and explicitly state—preferably at the very beginning of the manuscript—the precise extent to which this interpretation holds.

  2. Related to the point above, referring to the quadruple $(G, K, \omega_{D+1}, \alpha_D)$ as a “symmetry” can be somewhat confusing. The “fractional topological response” discussed in the manuscript appears to characterize the phase rather than the symmetry itself. Specifically, in the section “Description using bulk TQFT”, it is unclear how the fractional topological response—associated with $\alpha_D \in C^D(K,U(1))$—is encoded within the bulk TQFT data. The information contained in the higher fusion category, or equivalently the bulk TQFT together with a choice of gapped boundary condition, should only be sensitive to the cohomology class $[\omega_{D+1}]$ of the cocycle $\omega_{D+1}$ and an element of $H^{D+1}(K,U(1))$, rather than the specific cocycle $\alpha_D \in C^D(K,U(1))$. For example, in the $D=2$ case studied in Ref. [31], the data of the fusion category is specified by $(G, K, [\omega_3], [\alpha_2])$, depending only on the cohomology classes of the cocycles.

Additional Minor Comments:

  1. It is important throughout the manuscript to maintain a clear distinction between cocycles and their cohomology classes. For clarity and precision, consistent notation should be adopted—for example, using $[\omega_{D+1}] \in H^{D+1}(G, U(1))$ for a cohomology class and $\omega_{D+1} \in Z^{D+1}(G, U(1))$ for a representative cocycle. The current usage is inconsistent and potentially misleading.

  2. The term “generalized Frobenius-Schur indicator” is used ambiguously. The Frobenius-Schur indicator is an invariant associated with self-dual objects in fusion categories and should not be conflated with the full associator.

  3. Equation (2.5) is difficult to parse. Each symbol and component should be properly defined before use. In particular, the notation $i^A$ lacks a clear definition—only some of its properties are mentioned.

Recommendation

Ask for major revision

  • validity: high
  • significance: high
  • originality: good
  • clarity: ok
  • formatting: reasonable
  • grammar: -

Author:  Ryohei Kobayashi  on 2025-11-05  [id 5995]

(in reply to Report 2 on 2025-06-09)

  1. The manuscript suggests that coset symmetry arises from startin with a global symetry group $G$ possessing a 't Hooft anomaly classified by $[\omega_{D+1}]\in H^{D+1}(G,U(1))$ and subsequently gauging an anomaly-free subgroup $K\subset G$ with a discrete torsion inside $H^D(K,U(1))$. The authors need to clearly and explicitly state -- preferably at the very beginning of the manuscript -- the precise extent to which this interpretation holds. Reply: This is actually how we define coset symmetry. We have edited section 1.2 Summary of Results to state that ``Such symmetry is related to invertible $G$ symmetry by gauging a $K$ subgroup symmetry.''

  2. Related to the point above, referring to the quadruple $(G,K,\omega_{D+1},\alpha_D)$ as a "symmetry" can be somewhat confusing. The "fractional topological response" discussed in the manuscript appears to characterize the phase rather than the symmetry itself. Specifically, in the section "Description using bulk TQFT," it is unclear how the fractional topological response -- associated with $\alpha_D\in C^D(K,U(1))$-- is encoded within the bulk TQFT data. The information contained in the higher fusion category, or equivalently the bulk TQFT together with a choice of gapped boundary condition, should only be sensitive to the cohomology class $[\omega_{D+1}]$ of the cocycle $\omega_{D+1}$ and an element of $H^{D+1}(K,U(1))$ rather than the specific cocycle $\alpha_D\in C^D(K,U(1))$. For example, in the $D=2$ case studied in Ref.[31], the data of the fusion category is specified by $(G,K,[\omega_3],[\alpha_2])$ depending only on the cohomology classes of the cocycles. Reply: We include the fractional topological response as part of the symmetry data similar to how systems with Lieb-Schultz-Mattis (LSM) anomalies from translation and filling constraints, one also has to specify the filling. The filling is not a piece of data from the symmetry algebra and indeed does not appear in the symTFT construction, but can still impose important constraints on dynamics. It would be interesting to understand how to incorporate this data into the symTFT description.

  3. It is important throughout the manuscript to maintain a clear distinction between cocycles and their cohomology classes. For clarity and precision, consistent notation should be adopted—for example, using $[\omega_{D+1}]\in H^{D+1}(G,U(1))$ for cohomology class and $\omega_{D+1}\in Z^{D+1}(G,U(1))$ for a representative cocycle. The current usage is inconsistent and potentially misleading. Reply: We thank the referee for pointing this out and have modified the text to specify cohomology class vs representative cocycle. For instance, we denote a cohomology class containing a cocycle representative $\omega$ by $[\omega]$.

  4. The term "generalized Frobenius-Schur indicator is used ambiguously. The Frobenius-Schur indicator is an invariant associated with self-dual objects in fusion categories and should not be conflated with the full associator. Reply: We do not use FS indicator to refer to the full associator. Rather, we use it to refer to a particular piece of data of the associator. In the case where the FS indicator $\pm 1$ describes a property of self-dual duality objects in a Tambara-Yamagami fusion category (i.e. Ising) based on an abelian group $G$, it has a simple interpretation in the corresponding symTFT. SymTFTs for different FS indicators can be obtained by gauging a $\mathbb{Z}_2$ symmetry of $G$ gauge theory, that may permute anyon types, with different 2+1d $\mathbb{Z}_2$ SPTs (in arXiv:1410.4540v4, stacking with different 2+1d SPTs lead to different defectification classes). For example, the symTFT of such a fusion category with a $-1$ FS indicator can be obtained from gauging a $\mathbb{Z}_2$ permutation symmetry $G$ gauge theory with a nontrivial $\mathbb{Z}_2$ SPT. This is discussed in detail in arXiv:2304.01262 for 1+1d (2+1d symTFT) and arXiv:2308.11706 for 3+1d (4+1d symTFT) for duality symmetries. In the latter example, the objects are not self dual but rather order four, and the classes of 4+1d $\mathbb{Z}_4$ SPTs are referred to as "generalized FS indicators." We use "generalized FS indicators" in this work in the same way: we use them to refer to different bulk SPTs that we can stack before gauging a diagonal symmetry. The resulting different SymTFTs correspond to symmetries with slightly modified associators. We have added some discussion of this definition of generalized FS indicator to the beginning of Sec. 2.4.

  5. Equation (2.5) is difficult to parse. Each symbol and component should be properly defined before use. In particular, the notation $i^A$ lacks a clear definition -- only some of its properties are mentioned. Reply: We thank the referee for pointing this out. We added a few paragraphs to this section to clarify the definitions of slant products $i^A, i^B$ in Eq. 2.6 and 2.7.

Attachment:

reply.pdf

---

## Round 2 · Referee Report · Anonymous (Referee 2) · 2025-11-7

Report

I thank the authors for their detailed response to my comments. I am satisfied with the revisions, and I recommend the manuscript for publication in its current form.

Recommendation

Publish (surpasses expectations and criteria for this Journal; among top 10%)

---

## Round 2 · Referee Report · Anonymous (Referee 1) · 2025-11-10

Report

The authors have addressed all the comments in my previous report satisfactorily. I therefore recommend the revised paper for publication.

Recommendation

Publish (surpasses expectations and criteria for this Journal; among top 10%)

---

## Round 2 · Author Response

We thank the referees for their valuable and constructive comments. Below, we provide detailed responses to each point. Corresponding revisions have been made in the manuscript, with newly added content highlighted in red.

---

## Round 2 · List of Changes

1. We further explained our definition of Frobenius Schur indicator at the beginning of Sec.2.4.
  2. In Eq.2.6 and Eq.2.7 of the revised manuscript, we added the definitions of the slant products $i_g^A, i_g^B$.
  3. In Sec.3.6, we found that the statement of symmetry-enforced gaplessness should be separated into spacetime dimensions $D\ge 3$ and $D=2$; when $D=3$ the statement is shown in fully general, while at $D=2$ the argument is valid only for $G/K$ such that $H^2(BK',U(1))=0$ with any subgroup $K'\subset K$. We made edits of Sec.3.6 accordingly.
  4. We have replaced an example of a gapless phase with twisted coset symmetry in Sec.2.6.2.

---

## Editorial Decision

voting_in_preparation